# A robust cis-Mendelian randomization method with application to drug target discovery

Zhaotong Lin [1,2] ✉ & Wei Pan [1]

Mendelian randomization (MR) uses genetic variants as instrumental variables (IVs) to investigate causal relationships between traits. Unlike conventional MR, *cis*-MR focuses on a single genomic region using only *cis*-SNPs. For example, using *cis*-pQTLs for a protein as exposure for a disease opens a cost-effective path for drug target discovery. However, few methods effectively handle pleiotropy and linkage disequilibrium (LD) of *cis*-SNPs. Here, we propose cisMR-cML, a method based on constrained maximum likelihood, robust to IV assumption violations with strong theoretical support. We further clarify the severe but largely neglected consequences of the current practice of modeling marginal, instead of conditional genetic effects, and only using exposure-associated SNPs in *cis*-MR analysis. Numerical studies demonstrated our method's superiority over other existing methods. In a drug-target analysis for coronary artery disease (CAD), including a proteome-wide application, we identified three potential drug targets, PCSK9, COLEC11 and FGFR1 for CAD.

Mendelian randomization is a widely-used approach that uses genetic variants as instrumental variables (IVs) to infer the causal relationship between a pair of traits, one called exposure and another outcome. Since genetic variants are randomly allocated and fixed at conception, it reduces the risk of confounding and reverse causation with observational data[1,2]. Within the IV regression framework, MR requires three valid IV assumptions for valid inference: the IVs must be (1) associated with the exposure; (2) independent of any confounders of the exposure-outcome relationship; (3) not associated with the outcome conditional on the exposure and confounders. Subject to these assumptions, MR can provide evidence of a (putative) causal relationship between the exposure and the outcome, and the inverse variance weighting (IVW) method[3] can be applied. However, only the first IV assumption can be tested and is relatively easy to be satisfied in practice by using genome-wide significant SNPs associated with the exposure; in contrast, the second and third assumptions cannot be tested empirically and are likely to be violated due to the presence of wide-spread (horizontal) pleiotropy. Numerous MR methods have been proposed in the likely presence of horizontal pleiotropy[4–8], but

most of them require the use of independent IVs as conducted in most MR analyzes.

Meanwhile, there has been a growing interest in MR studies focusing on a small genomic region using some local and correlated *cis*-SNPs as IVs, known as *cis*-MR. One of the most promising applications of *cis*-MR is for drug target discovery, including drug target prioritization, validation or repositioning[9–11]. A drug-target MR analysis uses a protein (as a potential drug target) or its downstream biomarker as the exposure, and corresponding *cis*-SNPs of the gene encoding the protein as IVs. Despite the significance of such an analysis, it still depends crucially on the three valid IV assumptions. While using proteins as exposures makes it less likely to violate the assumption of no horizontal pleiotropy, as proteins are causally upstream of many common risk factors used in traditional/polygenic MR[9,12], different biological mechanisms may still exist even among the *cis*-SNPs in the same gene/protein region. For example, genetic variation in a transcription factor (TF)-binding site will potentially influence the binding affinity or efficiency of the TF, which may subsequently affect the production of the associated RNA and proteins; given that most genes

[1]Division of Biostatistics and Health Data Science, University of Minnesota, Minneapolis, MN 55455, USA. [2]Present address: Department of Statistics, Florida State University, Tallahassee, FL 32306, USA. ✉e-mail: zl23k@fsu.edu

have multiple potential TF-binding sites, genetic variations in the *cis*-region of a gene may involve distinct biological mechanisms[13]. One can first perform linkage disequilibrium (LD) clumping to obtain some (approximately) independent IVs before applying one or more of the existing robust MR methods based on independent IVs, however, it would lead to possibly severe loss of power due to only one or few independent SNPs remaining[12]; in fact, with only one or two SNPs, many robust MR methods cannot be applied. Finally and more importantly, as will be shown in subsequent numerical studies, only using independent SNPs is highly likely to result in the absence of a valid IV in the analysis due to their correlations with other SNPs in the region. As an alternative, we would rather use multiple correlated IVs in *cis*-MR. However, only few *cis*-MR methods are robust to the violation of the IV assumptions. Perhaps the most widely used *cis*-MR method is the generalized MR-IVW[14], which uses generalized linear regression to account for LD (among correlated SNPs) but assumes all valid IVs. Similarly, a generalized version of MR-Egger[15] and another closely related method, LDA-Egger[16], require a stringent (so-called InSIDE) assumption on the relationship between the unknown IV strengths and pleiotropic effects; furthermore, more generally, MR-Egger is low powered and sensitive to the coding of the SNPs[17]. There are several recently proposed methods to account for both LD and horizontal pleiotropy, such as MR-LDP[18], MR-Corr2[19], MR-CUE[20], MRAID[21] and RBMR[22]. All these methods impose different modeling assumptions on the distribution of the latent/hidden pleiotropic effects, while some can only handle either correlated pleiotropy or uncorrelated pleiotropy, but not both. Furthermore, these methods are proposed in the context of using a complex trait as the exposure (or in a polygenic MR setup), including a relatively large number of SNPs across the whole-genome as IVs, and may not be most suitable for *cis*-MR applications, as will be demonstrated later in our simulation studies.

In this work, we propose a robust *cis*-MR method called cisMR-cML, extending MR-cML[7] to allow for correlated SNPs as IVs. As its previous version with independent IVs, cisMR-cML is robust to violation of any one, two or all three IV assumptions, imposing minimum modeling assumptions with strong theoretical support. Furthermore, we clarify two main differences between cisMR-cML and MR-cML. First, in cisMR-cML we model conditional/joint SNP effects, instead of marginal effects as directly taken from GWAS summary data. Second, when selecting SNPs as IVs in the analysis, we include not only SNPs associated with the exposure, but also those associated with the outcome. These two differences are important: due to the use of correlated SNPs, failing to do so may lead to using all invalid IVs. These two differences have been largely neglected in the literature, but have severe consequences for any extension of other robust MR methods to *cis*-MR with correlated IVs, such as the median-based[4] and mode-based[23] MR methods. We show the robustness of the proposed method to the presence of invalid IVs in simulation studies, and illustrate the severe consequence of using only SNPs that are (conditionally) associated with the exposure. Lastly, we demonstrate the effectiveness of the proposed method in two real data applications for drug target discovery for coronary artery disease (CAD). In the first application, we use downstream biomarkers to serve as a proxy for the perturbation of a drug target, while in the second one, we perform a proteome-wide analysis to identify some proteins as potential drug targets for CAD.

## Results

### Overview of the proposed cisMR-cML method

We propose cisMR-cML to estimate the causal effect of an exposure (e.g. a gene or a protein) on an outcome in the possible presence of invalid IVs using publicly available GWAS summary data. It is an important extension of MR-cML[7] to allowing for correlated SNPs as IVs encountered with a molecular exposure such as a gene or a protein. By allowing for the use of correlated IVs, cisMR-cML is suitable for *cis*-MR

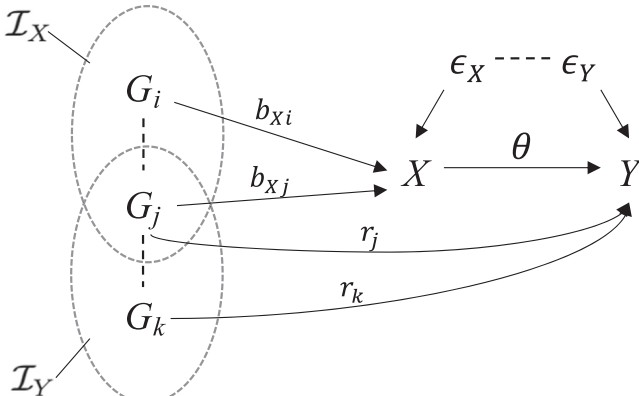

**Fig. 1 | Causal diagram among multiple SNPs ($G$), the exposure ($X$) and the outcome ($Y$).** $\epsilon_X$ and $\epsilon_Y$ are random noises, $\theta$ is the causal effect of interest, $b_{Xi}$'s are conditional effects of SNPs on the exposure, $r_i$'s are horizontal pleiotropic effects.

analysis, while it may not be feasible to have at least three independent IVs within a *cis*-region for the use of MR-cML. Additionally, it has two key distinctions from MR-cML that enhance its robustness to invalid IVs. First, as depicted in the general causal model Fig. 1, cisMR-cML models the conditional effects between SNPs and the exposure, and those between SNPs and the outcome, which differs from the conventional approach of modeling marginal GWAS estimates in MR-cML and other MR methods. This distinction significantly mitigates the risk of introducing additional (and unnecessary) horizontal pleiotropy when dealing with correlated SNPs in *cis*-MR (see Modeling conditional effects versus marginal effects). Second, unlike the usual practice of only using SNPs relevant to the exposure, cisMR-cML uses variants that are jointly associated with either the exposure or the outcome as IVs, i.e., variants in $\mathcal{I}_X \cup \mathcal{I}_Y$. And we use a conditional and joint association analysis called GCTA-COJO[24] to select these variants. Properly accounting for outcome-associated SNPs further helps to avoid additional horizontal pleiotropy (see Selection of genetic variants as IVs in cisMR-cML). Although these two essential considerations proposed in cisMR-cML may seem elementary statistically, they are often overlooked in current *cis*-MR applications. For example, two most widely-used *cis*-MR methods, generalized IVW and generalized Egger (see Methods), directly model the marginal GWAS estimates; and the recent two drug-target applications (e.g. Zhao et al.[10]; Zheng et al.[25]) applied them followed by selecting conditionally independent pQTLs.

Once the genetic variants as candidate IVs are chosen, the LD matrix among these variants can be estimated using a publicly available reference panel. The marginal estimates from GWAS summary data are then converted into conditional GWAS estimates. Then under the two-sample MR framework, cisMR-cML is implemented in a maximum likelihood framework under a constraint on the number of invalid IVs with horizontal (correlated and/or uncorrelated) pleiotropy; the number of invalid IVs is selected consistently by the Bayesian Information Criterion (BIC). In short, cisMR-cML selects valid IVs from the candidate set $\mathcal{I}_X \setminus \mathcal{I}_Y$ to infer the causal relationship from $X$ to $Y$. We establish statistical theory to demonstrate some desirable properties of cisMR-cML, such as its estimation consistency and asymptotic normality in the presence of invalid IVs with correlated or uncorrelated pleiotropy. Finally, we also implement a data perturbation (DP) approach to account for the uncertainty in model selection (see Methods).

### Simulations: cisMR-cML outperforms existing methods in the presence of invalid IVs

We conducted two sets of simulation studies to compare our proposed method (with data perturbation unless specified otherwise) and other existing *cis*-MR methods including generalized IVW and Egger (GIVW

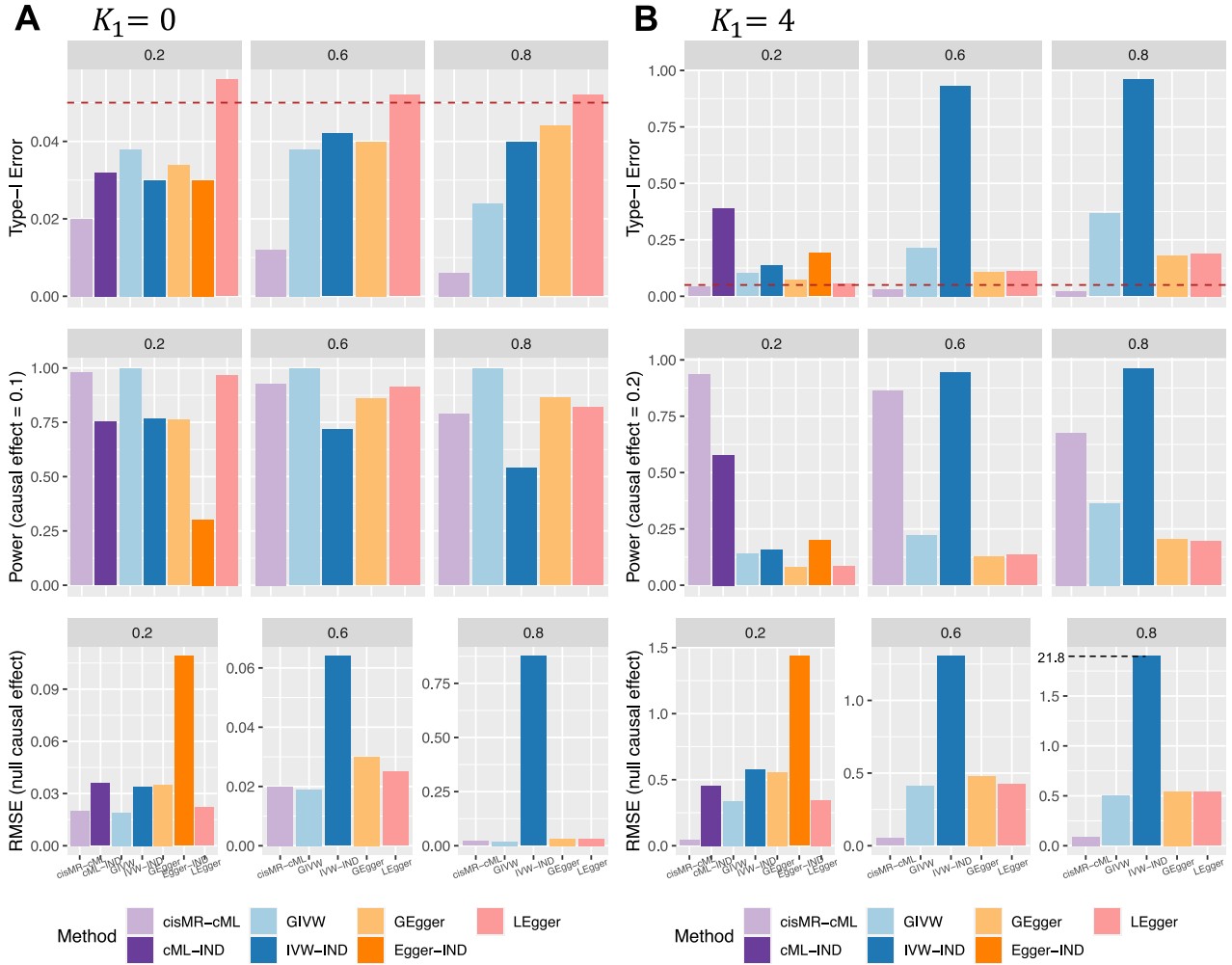

**Fig. 2 | Simulation results in scenario 1 with autoregressive LD pattern.** All 10 IVs have an effect on the exposure. **A** $K_1 = 0$ (no invalid IV). **B** $K_1 = 4$ invalid IVs. From top to bottom are empirical type-I error under the null, power under the alternative, and root mean squared error (RMSE) under the null. Colors represent different methods being compared. Source data are provided as a Source Data file.

and GEgger)[14], LD-aware Egger[16] (LEgger) (see Methods), and different implementations of these methods. In the first set of simulation studies, we directly generated GWAS summary statistics for 10 SNPs from an autoregressive LD pattern with a weak correlation ($\rho = 0.2$), a moderate correlation ($\rho = 0.6$), or a strong correlation ($\rho = 0.8$), and considered two scenarios: (1) all 10 SNPs had an effect on the exposure, i.e., $|\mathcal{I}_X| = 10$; (2) only half of the SNPs had an effect on the exposure, i.e., $|\mathcal{I}_X| = 5$, and $|\mathcal{I}_Y \setminus \mathcal{I}_X| = 2$. In both scenarios, we varied the number of invalid IVs in $\mathcal{I}_X \cap \mathcal{I}_Y$, denoted as $K_1$. We performed several methods: cisMR-cML and LEgger with the conditional estimates calculated based on all 10 SNPs; GIVW and GEgger with the marginal GWAS estimates. In scenario (1), we also selected independent IVs (with $r^2 < 0.001$) and applied the independent versions of IVW, Egger and MR-cML with marginal GWAS estimates. We referred to these implementations as IVW-IND, Egger-IND and cML-IND. We further applied four polygenic MR methods that can account for LD but were not specifically proposed for *cis*-MR analysis, including MR.LDP, MR.Corr2, MR.CUE and MRAID. In scenario (2), we additionally investigated the performance of different *cis*-MR methods using only SNPs in $\mathcal{I}_X$. Specifically, we applied cisMR-cML and LEgger with the conditional estimates calculated only based on the 5 SNPs in $\mathcal{I}_X$; and applied GIVW and GEgger with the GWAS summary data of the 5 SNPs in $\mathcal{I}_X$. We referred to these implementations as cisMR-cML-X, LEgger-X, GIVW-X and GEgger-X respectively.

In the first scenario, where all 10 IVs had effects on the exposure ($|\mathcal{I}_X| = 10$), representative results were presented in Fig. 2, and full results can be found in Supplementary Section S3.1. Throughout our simulations, type-I error was evaluated at the significance level of 5%. First, when all 10 IVs were valid (Fig. 2A and Supplementary Table S1), all methods yielded well-controlled type-I error rates. cisMR-cML (with the data perturbation implementation) was more conservative than other methods in this ideal scenario with no invalid IV, which was similar to what was observed before in MR-cML[7] and MVMR-cML[26]. It is also noted that, even in such an ideal scenario, GEgger had a relatively larger root mean squared error (and less precise estimates) than the other three *cis*-MR methods, which may be due to the allele orientation step implemented in the method (see Lin et al.[17] for more discussion on this issue). IVW-IND, Egger-IND and cML-IND all had lower power than their correlated version counterparts, namely GIVW, GEgger and cisMR-cML. In the presence of 4 invalid IVs (Fig. 2B), only cisMR-cML could control the type-I error and at the same time maintain high power. Furthermore, it had a much lower RMSE than the other three methods. On the other hand, GIVW, GEgger and LEgger had increasingly inflated type-I errors as the correlations among SNPs increased. The three approaches using independent IVs also had highly inflated type-I errors as the direct effects of invalid IVs were absorbed in the marginal GWAS effects used in the analysis due to LD. As shown in Supplementary Table S2, the four polygenic MR methods exhibited

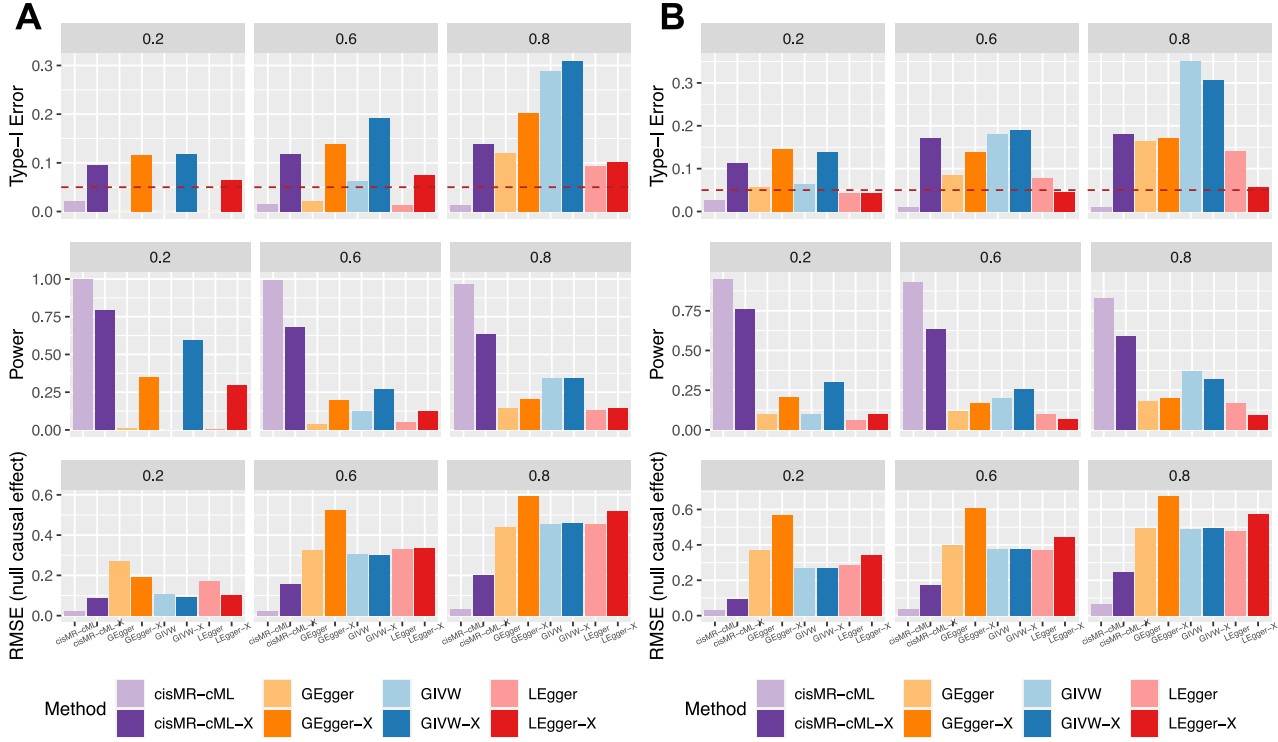

**Fig. 3 | Simulation results in scenario 2 with $|\mathcal{I}_X| = 5, |\mathcal{I}_Y \setminus \mathcal{I}_X| = 2$. A** $K_1 = |\mathcal{I}_X \cap \mathcal{I}_Y| = 0$. **B** $K_1 = |\mathcal{I}_X \cap \mathcal{I}_Y| = 1$. From top to bottom are empirical type-I error under the null, power under the alternative, and root mean squared error (RMSE) under the null. Colors represent different methods being compared. Source data are provided as a Source Data file.

unstable performance, with either extremely low power, inflated type-I errors, or unsuccessful convergence. Finally, we further investigated the use of marginal GWAS estimates in cisMR-cML as shown in Eq. (15) (referred to as cisMR-cML-Marg). As detailed in Supplementary Section S3.2, cisMR-cML-Marg had highly inflated type-I errors due to the violation of plurality assumption in the marginal model, as discussed in Modeling conditional effects versus marginal effects.

In the second scenario with only 5 SNPs having effects on the exposure, we further examined the performance of the four methods using only the data from these 5 SNPs, as illustrated by the suffix '-X' in Fig. 3. When $K_1 = 0$ (Fig. 3A, Supplementary Table S7), it seemed that all IVs in $\mathcal{I}_X$ were valid. However, due to their correlations with those in $\mathcal{I}_Y$, they absorbed the direct effects of the SNPs in $\mathcal{I}_Y$ on the outcome if we failed to include the SNPs in $\mathcal{I}_Y$. Therefore, all the IVs in $\mathcal{I}_X$ became invalid and the plurality condition was violated in cisMR-cML-X, which yielded highly inflated type-I errors. Similarly, GIVW-X and GEgger-X, only using SNPs conditionally associated with the exposure also yielded inflated type-I errors. On the other hand, cisMR-cML using all 10 SNPs yielded well-controlled type-I errors, high power and the smallest RMSE across all scenarios. Through this example, we can see the importance of including the SNPs in $\mathcal{I}_Y$ besides those in $\mathcal{I}_X$ when calculating the conditional estimates, because otherwise, the plurality condition required by cisMR-cML (or more generally by model identification) may be violated (unless there was no or little LD between the SNPs in $\mathcal{I}_X$ and $\mathcal{I}_Y$, which will be considered later). Additionally, it is worth mentioning that when $K_1 = 1$ in Fig. 3B (Supplementary Table S8), among the 10 IVs used in cisMR-cML, some violated only the 'relevance' assumption, some violated only the 'no horizontal pleiotropy' assumption, but some violated both assumptions. Notably, the proposed method performed robustly in the presence of different types of invalid IVs, producing unbiased estimates and well-controlled type-I errors.

In the second scenario, we also considered two other ways of selecting the set of SNPs to be used in cisMR-cML. Specifically, we compared mistakenly including some irrelevant IVs (i.e., the previous implementation of using all 10 SNPs including the three SNPs in $(\mathcal{I}_X \cup \mathcal{I}_Y)^C$) versus only using SNPs in $\mathcal{I}_X \cup \mathcal{I}_Y$. We also considered the situation where we failed to include some relevant SNPs in $\mathcal{I}_X$ by using $\mathcal{I}_{X_s} \cup \mathcal{I}_Y$, where $\mathcal{I}_{X_s} \subset \mathcal{I}_X$ and $|\mathcal{I}_{X_s}| = 3$. We found that although all three approaches yielded unbiased estimates and well-controlled type-I errors, using SNPs in $\mathcal{I}_X \cup \mathcal{I}_Y$ gave the highest power while using irrelevant SNPs or omitting some true (conditionally) relevant SNPs led to a loss of power. More details and results are provided in Section S3.5 in the Supplementary.

We further evaluated the performance of different approaches under a few more scenarios (see Simulations with autoregressive correlation structure for simulation details). First, when only weak IVs were available, which were simulated by reducing the genetic effect sizes on the exposure or the exposure GWAS sample size in scenario 1 with $K_1 = 0$, all methods could control type-I error. Particularly, only the proposed cisMR-cML yielded unbiased estimates, while other methods exhibited different degrees of bias (Supplementary Tables S10, 11). Second, we considered the scenario where the SNPs in $\mathcal{I}_Y$ were weakly invalid with pleiotropic effects $r_i = \kappa/\sqrt{N_Y}$, i.e., pleiotropic effects decreased as the outcome GWAS sample size increased at rate $1/\sqrt{N_Y}$. We varied the value of $\kappa$ and $N_Y$ while keeping $N_X$ the same. We observed that as $\kappa$ increased, cisMR-cML-BIC had improved performance in selecting out invalid IVs based on BIC. When $\kappa = 1$, cisMR-cML-BIC barely selected out any invalid IVs and thus had inflated type-I errors; however, cisMR-cML-DP (with data perturbation) still performed reasonably well with controlled type-I errors and nearly unbiased estimates (Table S12). When $\kappa = 5$, cisMR-cML-BIC oftentimes failed to select out all 4 invalid IVs, and cisMR-cML-DP also yielded biased estimates and slightly inflated type-I errors (Table S13). When $\kappa$

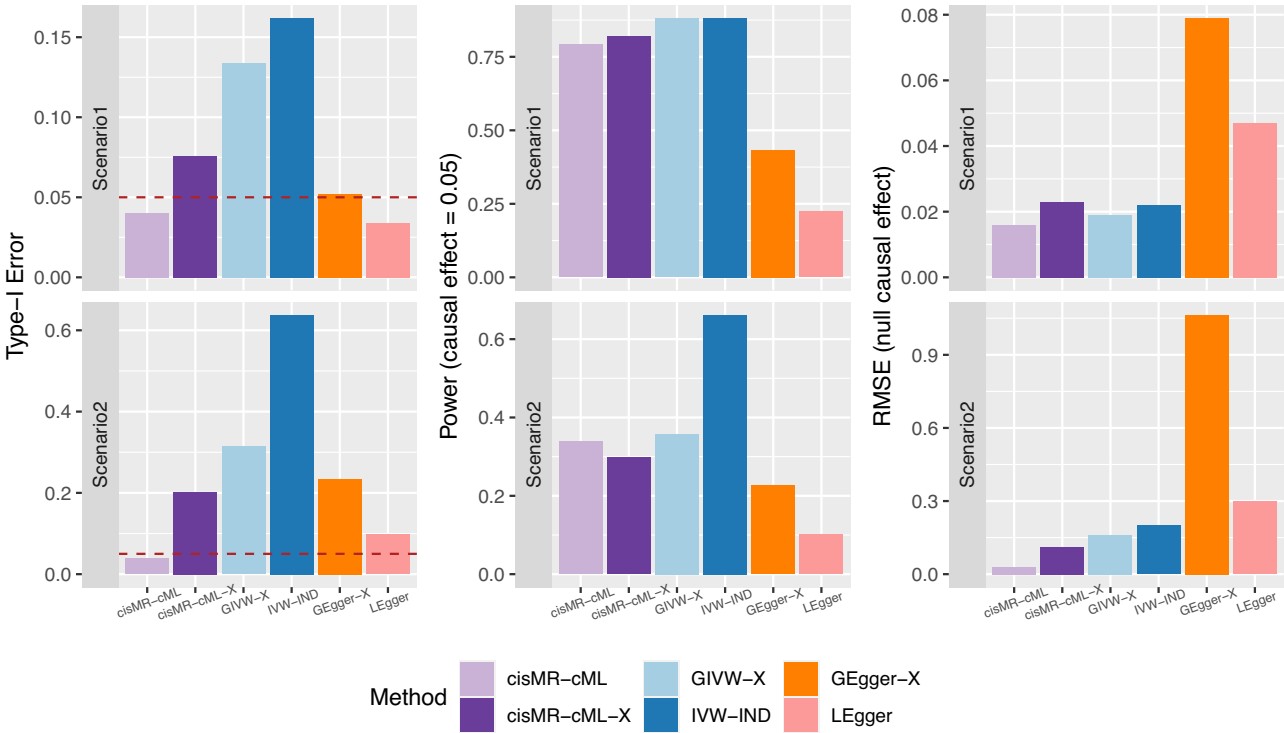

**Fig. 4 | Simulation results with real LD patterns derived from UK Biobank data.** Top: Scenario 1 with SNPs in $\mathcal{I}_X$ and $\mathcal{I}_Y$, and corresponding genetic effect sizes specified according to real data. Bottom: Scenario 2 with SNPs in $\mathcal{I}_X$ and $\mathcal{I}_Y$, and corresponding genetic effect sizes randomly generated. From left to right are empirical type-I error under the null, power under the alternative, and root mean squared error (RMSE) under the null. Colors represent different methods being compared. Source data are provided as a Source Data file.

became large ($\kappa = 20$), cisMR-cML-BIC often successfully identified all 4 invalid IVs and had well-controlled type-I errors (Table S14). We also explored the performance of cisMR-cML-BIC based on asymptotic inference when both $N_X$ and $N_Y$ increased and $r_i$ diminished at different rates. Results are discussed in Supplementary Section S1 and Tables S16–S18. Third, when SNPs in $\mathcal{I}_Y$ were not correlated with SNPs in $\mathcal{I}_X$, all examined approaches, including cisMR-cML, cisMR-cML-X, GIVW-X, GEgger-X and LEgger-X, performed well. This was expected since no pleiotropy was introduced via LD and the SNPs in $\mathcal{I}_X$ were still valid IVs. However, in the case that the SNPs in $\mathcal{I}_Y$ were weakly correlated with the SNPs in $\mathcal{I}_X$, cisMR-cML-X and GIVW-X had highly inflated type-I errors, which again highlighted the importance of including the SNPs in $\mathcal{I}_Y$ in the *cis*-MR analysis (Supplementary Table S15). Detailed results for the above three additional scenarios are provided in Supplementary Section S3.6.

Next, we turned to the second set of simulation studies, in which more realistic LD patterns were examined by generating data based on our real data analysis. Briefly, we started by generating the exposure and outcome data from the UK Biobank individual genotypes in the *cis*-region of a given gene/protein, then performed exposure and outcome GWAS on two non-overlapping samples. We then applied GCTA-COJO on the exposure and the outcome GWAS to select SNPs jointly associated with $X$ (denoted as $\mathcal{C}_X$) and jointly associated with $Y$ (denoted as $\mathcal{C}_Y$) respectively, using a third non-overlapping sample as the reference LD panel. We considered two scenarios: in scenario 1, 50 proteins with at least five SNPs associated with the exposure and one SNP associated with the outcome based on the COJO analysis in the proteome-wide application were randomly selected. Then we specified SNPs in $\mathcal{I}_X$ and $\mathcal{I}_Y$ and corresponding genetic effect sizes exactly based on the real-data COJO results and real pQTL/GWAS data. In scenario 2, 50 proteins were randomly selected, and SNPs in $\mathcal{I}_X$ and $\mathcal{I}_Y$ and their corresponding genetic effect sizes were generated randomly. See Simulations with real LD patterns derived from UK Biobank data for data-generating details.

We evaluated the performance of cisMR-cML and LEgger using the SNPs in $\mathcal{C}_X \cup \mathcal{C}_Y$ selected by COJO based on the simulated data, and the common practice with cisMR-cML-X, GIVW-X and GEgger-X only using the SNPs in $\mathcal{C}_X$ for *cis*-MR analysis. We applied IVW-IND with the set of independent SNPs marginally associated with the exposure after LD clumping, as well as the four polygenic MR methods (MR.LDP, MR.Corr2, MR.CUE and MRAID) with the set of correlated SNPs marginally associated with the exposure (See Simulations with real LD patterns derived from UK Biobank data for implementation details). Figure 4 shows the results for 500 replicates (10 replicates per gene) in scenarios 1 (top row) and 2 (bottom row) respectively. Among the methods assessed, cisMR-cML was the only method that effectively controlled the type-I error and yielded the smallest RMSE. In contrast, LEgger demonstrated generally low power, and other approaches using only SNPs associated with the exposure in the analysis had unsatisfactory performance with inflated type-I errors. See Supplementary Table S19 for full results of all examined methods.

## Causal effects of downstream biomarkers on CAD
In the first real data application, we applied *cis*-MR in a setup where we used a downstream biomarker as a proxy of protein concentration and activity (see Drug-target MR application with the use of downstream biomarkers). Our analysis here mainly aims to illustrate how to apply cisMR-cML using a downstream biomarker of the target protein to confirm/replicate some well-established results.

Specifically, we assessed the causal relationship of low-density lipoprotein cholesterol (LDL) on coronary artery disease (CAD) using the genetic variants restricted to the *PCSK9* region. PCSK9 can bind to and break down LDL receptors, therefore decreasing the clearance of LDL cholesterol. PCSK9 inhibitors are a new type of drug that can lower LDL levels by blocking PCSK9 protein from breaking down LDL

receptors. The causal effect of LDL on CAD has been extensively studied by randomized trials and MR[27]. In particular, Ference et al.[28] found a protective effect on CAD of lowering LDL using a weighted PCSK9 genetic score to mimic the effect of PCSK9 inhibitor. In our analysis of LDL and CAD, GCTA-COJO selected 9 SNPs located in the *PCSK9* region, 8 of which were associated with LDL, and one was associated with CAD. Both cisMR-cML, LEgger, GIVW-X and GEgger-X suggested a significant positive causal effect of LDL on CAD risk, with p-values $7.3 \times 10^{-4}$, $9.2 \times 10^{-3}$, $8.6 \times 10^{-8}$, 0.02 respectively.

Following Gkatzionis et al.[11], we also assessed the causal relationship of testosterone level on CAD using the genetic variants in the *SHBG* region. While an association between low testosterone level and CAD risk has been reported in some observational studies, its causal relationship is still unclear. Sex hormone-binding globulin (SHBG) can bind to sex hormones in the blood and help control the amount of sex hormones. Multiple variants in this region have been demonstrated to be associated with testosterone. In the analysis of testosterone level and CAD, GCTA-COJO selected 14 SNPs associated with testosterone, and no SNP associated with CAD. Using the 14 variants in the *SHBG* region, no method identified any significant causal effect of testosterone on CAD risk, which was consistent with previous findings in Burgess et al.[12]; Schooling et al.[29]; Gkatzionis et al.[11].

## Proteome-wide analysis for CAD risk

In the second real data application, we used protein expression data as the exposure, which was a more direct proxy of the drug target, and we assessed their causal effects on the risk of CAD. Specifically, we did a proteome-wide scan using the pQTL summary data derived from ARIC European ancestry (EA) cohort with sample size $N_X = 7213$[30]. After data preprocessing (see A proteome-wide application to CAD), in total 773 proteins were analyzed. Among the 773 proteins, 183 proteins had at least one SNP in the *cis*-region associated with the outcome according to the COJO analysis (i.e., $|\mathcal{C}_Y \setminus \mathcal{C}_X| \geq 1$), and cisMR-cML-BIC detected over 98.7% of such invalid IVs. Furthermore, 30 protein-CAD pairs had one invalid IV in $\mathcal{C}_Y \cap \mathcal{C}_X$ detected by cisMR-cML-BIC. It is also noted that, with a finite sample size, the selection of invalid IVs based on BIC may not be perfect, especially in the presence of weak invalid IVs. With data perturbation, cisMR-cML has detected 310 protein-CAD pairs with at least one invalid IV in $\mathcal{C}_Y \cap \mathcal{C}_X$ over 10% of the time during the 100 data perturbations.

We used the Benjamini-Hochberg approach to account for multiple testing in our proteome-wide analysis, and reported significant MR findings with a false discovery rate (FDR) less than 0.05. We also conducted colocalization analysis on the significant proteins with an FDR-adjusted p-value less than 0.05 (see Methods). cisMR-cML identified three proteins with putative causal effect on CAD risk, including PCSK9, COLEC11 and FGFR1. Using a threshold of H4-PP ≥0.7, there was colocalization evidence for both PCSK9 and COLEC11. As discussed in the previous application, PCSK9 inhibitors can lower LDL levels, which is a major risk factor for CAD. Several trials found that evolocumab, a PCSK9 inhibitor, can significantly lower LDL levels and cardiovascular disease risk[31–33]. COLEC11 is involved in lectin complement activation pathway and plays an important role in the innate immune system. The vital role of the complement system in heart diseases has been studied, including promoting inflammation, tissue damage, etc.[34,35]. While complement inhibitors have been suggested as a potential therapeutic target for heart disease, more studies on the relationship between COLEC11 and CAD are warranted. As for FGFR1, colocalization only identified the causal variant for the protein with H1-PP ≈ 96%. This was the scenario with insufficient evidence for association with CAD in the CAD GWAS data[36]. FGF/FGFR signalling plays an important role in cell proliferation and angiogenesis, and several FGFR1 inhibitors have been used to treat various types of cancer[37]. While overexpression of FGFR1 may play a role in the development of cardiac hypertrophy[38,39], it is also likely that FGFR expression pattern is altered in response to cardiac stress and injury and facilitate cardiac remolding[40,41]. Further studies are needed to fully understand their complex relationship.

On the other hand, GIVW-X identified 18 proteins, and four of them had colocalization evidence, including BMP1, COLEC11, PCSK9, and ERAP2. However, there were five proteins with an H3-PP greater than 0.7, including SWAP70, HTRA1, CXCL12, PDE5A, and ITIH3, which suggested that each protein and CAD might have distinct causal variants that were in linkage disequilibrium, and thus MR assumptions may be violated. In fact, for all these five proteins, COJO selected at least one SNP associated with CAD in the corresponding *cis*-region. cisMR-cML yielded non-significant p-values for four of them, except for PDE5A, which had a marginally significant p-value of 0.003 by cisMR-cML. Interestingly, PDE5 inhibitors have been found in some animal studies to have a potential cardioprotective effect[42], and a recent transcriptome-wide association analysis (TWAS) has also identified a positive effect of *PDE5A* on CAD in the aorta tissue[43]. However, in our proteome-wide analysis, both GIVW-X and cisMR-cML suggested a negative effect size of PDE5A on CAD. We further found that the QTL pairs, eQTLs from the GTEx (V8) aorta tissue and pQTLs from plasma used in our analysis, had opposite directions of effects. Such a discordance has been systematically investigated in Robinson et al.[44], which may be partially due to the difference in tissues. Nonetheless, such discordance may also be informative in drug target validation and the mechanism of PDE5 inhibitors on CAD is worth more investigation. Similarly, GEgger-X identified seven proteins, 2 of which had colocalization evidence including PCSK9 and TIRAP, and two had an H3-PP greater than 0.7. We note that this could be the scenario we've seen in our simulation studies, where only using SNPs conditionally associated with the exposure yielded inflate type-I error in GIVW-X and GEgger-X. Such significant MR findings may be attributable to genetic confounding through a variant in linkage disequilibrium as suggested by a high H3-PP. Wald-ratio test using the most significant pQTL identified 24 proteins, 4 of which had colocalization evidence, including PCSK9, TIRAP, COLEC11, and ERAP2, but 10 of them had evidence of H3-PP greater than 0.7. And lastly, LEgger didn't identify any significant results. We show the Q-Q plots of all methods in Fig. 5, in which we can see that in the left tail, cisMR-cML and LEgger had good alignment with the identity line, while GIVW-X, GEgger-X and Wald-ratio were inflated. The inflation factor for cisMR-cML was 1.00 (rounded to the second decimal), suggesting that the Type-I error was controlled satisfactorily; while LEgger, GEgger-X, GIVW-X and Wald-ratio test yielded inflation factors 0.95, 1.14, 1.57 and 1.54 respectively.

Finally, as a sensitivity analysis, we performed the analysis using different p-value thresholds in COJO to select IVs. As shown in the Supplementary, using $5 \times 10^{-8}$ led to a slightly inflated inflation factor of 1.08 for cisMR-cML, which may be partially due to the omission of SNPs in $\mathcal{I}_Y$, potentially causing cisMR-cML to violate the plurality condition as discussed in Selection of genetic variants as IVs in cisMR-cML. On the other hand, using some loose thresholds resulted in deflated values of inflation factor in the Q-Q plot, indicating that the proposed method may suffer from a loss of power, which could be partially due to the inclusion of many irrelevant (and highly correlated) SNPs. Therefore, as a trade-off we recommend $5 \times 10^{-6}$ as a starting point in COJO to select SNPs to be used in cisMR-cML. Lastly, regardless of the threshold used, alternative methods that utilized the marginal effect estimates of SNPs in $\mathcal{C}_X$, i.e., GIVW-X, GEgger-X, and Wald-ratio test, all yielded inflations.

## Discussion

We have proposed a robust *cis*-MR method called cisMR-cML, which uses correlated SNPs in a local genomic region to infer the causal relationship between a molecular exposure (e.g. a protein) and an outcome (e.g. CAD), and is robust to invalid IVs. It is an important extension of the existing MR-cML method, which has been shown to have good performance in practice but requires the use of

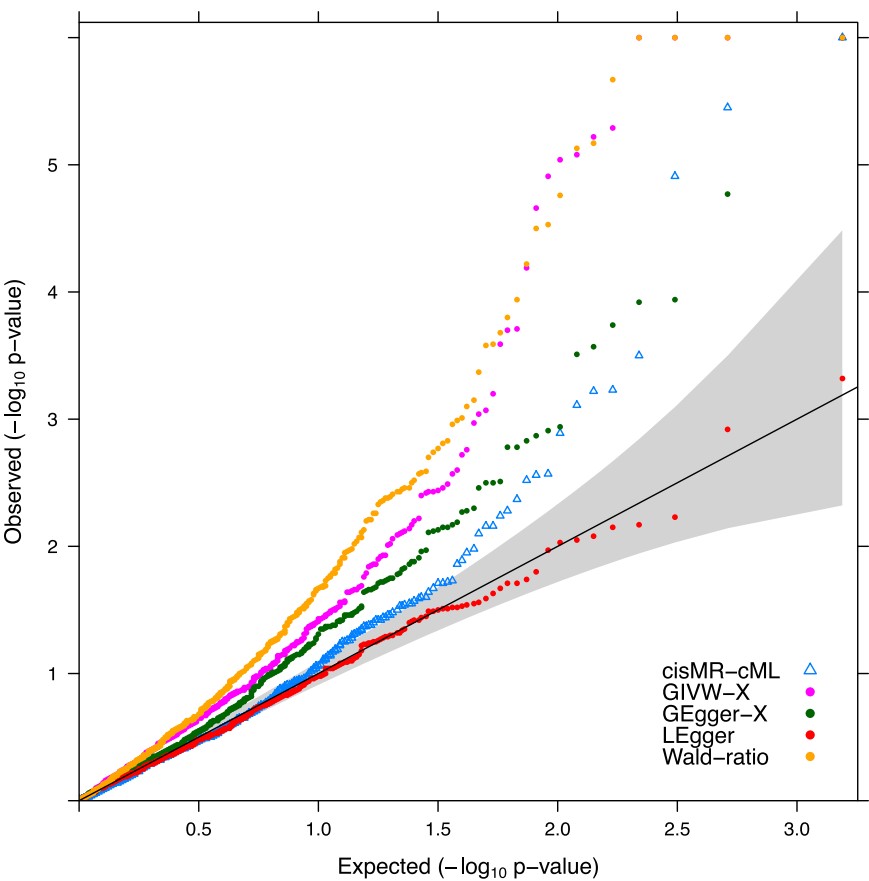

**Fig. 5 | Q-Q plots of $-\log_{10}$ p-value for different methods.** Colors represent the five different MR methods being compared. *p*-values are truncated at $1 \times 10^{-6}$ in this figure. The *p*-values are calculated based on corresponding two-sided MR tests without multiple testing adjustment. Source data are provided as a Source Data file.

independent SNPs[7]. While such an extension may seem straightforward at first glance by incorporating LD information in the likelihood, we have pointed out several important implementation details with significant implications for final results. To prevent inducing pleiotropy via LD, we adopted conditional association estimates, instead of marginal estimates, by suitably transforming GWAS summary data in cisMR-cML. We also discussed and demonstrated the importance of using SNPs associated with the outcome, in addition to those associated with the exposure, in cisMR-cML, which is in stark contrast to the common practice of only using SNPs associated with the exposure in MR, e.g. MR-cML. These caveats are expected to be applicable to the extensions of other existing robust MR methods only requiring the majority or plurality condition, such as weighted-median and mode-based methods. While we have mainly focused on the application of *cis*-MR using one genomic region in this paper, our method can be generalized to multiple independent LD blocks and serve as a useful MR method accounting for both LD and horizontal pleiotropy.

In our simulation studies, we have showcased much better performance of the proposed cisMR-cML over several commonly used *cis*-MR methods, including generalized IVW and Egger, and LDA-Egger. We have also compared different choices of the set of SNPs used in different methods. In particular, we have found that applying generalized IVW and Egger with only SNPs conditionally associated with the exposure may yield false positive findings, partly due to that some outcome-associated SNPs not included in the model are in LD with the SNPs in the model, thus leading to the violation of no-pleiotropy assumption. This was confirmed in both simulation studies and real data applications, and we hope to raise attention to this largely neglected issue in future applications.

We note some unique challenges with *cis*-MR (as compared to the standard polygenic MR) due to the use of correlated SNPs in the same cis-region as IVs. First, if an invalid IV is not included in the model, its correlations with all other IVs being included in the model would induce all other IVs being invalid. Second, if there is only one causal SNP and if it is already included as an IV, we would have only one valid IV. In both situations, the plurality condition will be violated. Our proposed approach of selecting and including any outcome-associated SNPs in the model while conducting model selection and using conditional effect estimates would alleviate, but not necessarily eliminate, the problem.

There are several limitations in this work. First, while cisMR-cML imposes minimum modeling assumptions, especially no additional assumption on the distribution of pleiotropic effects (except that their sizes are in the order of $O(1)$), it depends critically on the plurality condition, which depends on which SNPs are used in the model due to the correlations among all the SNPs, either selected or not, in a local region. We currently use GCTA-COJO to select and include the SNPs that are associated with either the exposure or the outcome, which has been previously found to perform better than p-value clumping[45]. Moreover, clumping is based on the marginal genetic association. In *cis*-MR, many marginal signals may stem from the same SNP, and when modeled in the conditional framework, their effect sizes may shrink to zero, potentially introducing more noises into the proposed approach. However, COJO is by no means the only method for conditional analysis. SNPs selection in *cis*-MR analysis is still an ongoing research topic (see Gkatzionis et al.[11] for a detailed review, and Schmidt et al.[9] for another example); there seems no consensus yet. How to incorporate other robust SNP selection techniques or develop new ones in cisMR-cML (or similar extensions) is of interest for future work. Moreover,

while the current algorithm works reasonably well in our experience, an algorithm with global optimization is desired to bridge the gap between the implementation and the theory. We also observed similarly conservative performance of cisMR-cML with the data perturbation implementation as its independent-IV (MR-cML[7,46]) and multivariable MR (MVMR-cML[26]) versions, which should be investigated further in the future. Second, since individual-level genotypes in the exposure and outcome GWAS data are often unavailable, as in most applications, we propose using a reference panel of similar ancestry to approximate an LD matrix. Such an approximation is known to introduce extra variation that is not taken into account in our method (and almost all methods); using a larger reference panel, such as the UK Biobank samples as used in our analysis, is expected to alleviate the problem[47]. Third, we have considered only the two-sample design with two independent GWAS datasets for the exposure and outcome. To account for overlapping samples between the two GWAS datasets, we may model the exposure estimates and the outcome estimates jointly with a multivariate normal distribution, instead of treating them as independent. Fourth, the proteome-wide application presented in this work is based on the protein levels measured in plasma samples, while using disease-relevant tissue samples may be preferred in drug-target MR. Since large-scale tissue-specific pQTL data are not yet available, one alternative is to use tissue-specific eQTL data as a proxy[9]. Relatedly, as shown in our example for protein PDE5A, discordance between eQTL data and pQTL data could happen, and only using one type of molecular data may yield misleading results. It is therefore recommended to combine and consolidate evidence from eQTLs and pQTLs for drug target validation[44]. Finally and importantly, triangulation with evidence from applying different *cis*-MR methods and colocalization analysis to observational data[36], and direct experimental studies when possible are warranted for more reliable causal inference.

## Methods

### Model

Based on Fig. 1 with $m$ SNPs ($G_1,...,G_m$), assuming that both the genotypes and the traits have been mean centered and standardized, the true models for the exposure $X$ and the outcome $Y$ are

$$\mathbf{X} = \sum_{i=1}^{m} b_{Xi}\mathbf{G}_i + \boldsymbol{\epsilon}_X, \qquad (1)$$

$$\mathbf{Y} = \theta\mathbf{X} + \sum_{i=1}^{m} r_i\mathbf{G}_i + \boldsymbol{\epsilon}_Y, \qquad (2)$$

where $\boldsymbol{\epsilon}_X$ and $\boldsymbol{\epsilon}_Y$ are random error terms independent of SNPs $\{\mathbf{G}_i\}_{i=1}^{m}$. In general, $\boldsymbol{\epsilon}_X$ and $\boldsymbol{\epsilon}_Y$ are correlated due to the presence of hidden confounding. Plugging Eq. (1) in Eq. (2), we have

$$\mathbf{Y} = \sum_{i=1}^{m} (\theta b_{Xi} + r_i)\mathbf{G}_i + (\theta\boldsymbol{\epsilon}_X + \boldsymbol{\epsilon}_Y) := \sum_{i=1}^{m} b_{Yi}\mathbf{G}_i + \boldsymbol{\epsilon}_Y^*, \qquad (3)$$

$$b_{Yi} = \theta b_{Xi} + r_i, \qquad (4)$$

where $\{b_{Xi}\}_{i=1}^{m}$ and $\{b_{Yi}\}_{i=1}^{m}$ are the joint/conditional effects of the $m$ SNPs on the exposure and the outcome respectively. Note that in Fig. 1, for SNPs $i \in \mathcal{I}_X$, we have $b_{Xi} \neq 0$; for SNPs $i \in \mathcal{I}_Y$, we have $r_i \neq 0$. Based on the three valid IV assumptions, a valid IV is a SNP with $b_{Xi} \neq 0$ and $r_i = 0$, i.e. $i \in \mathcal{I}_X \setminus \mathcal{I}_Y$. As discussed in Theorem 1 in Guo et al.[48], Eq. (4) is identifiable if and only if the valid IVs form the largest group of IVs sharing the same causal parameter value (i.e., the plurality condition).

Eqs. (1) and (2) are joint models for conditional association of $m$ SNPs on the exposure and on the outcome respectively, while typically in GWAS, the marginal associations of each SNP with the exposure and

with the outcome are modeled:

$$\begin{aligned}\mathbf{X} &= b_{Xi}^*\mathbf{G}_i + \boldsymbol{\epsilon}_X^*,\\ \mathbf{Y} &= b_{Yi}^*\mathbf{G}_i + \boldsymbol{\epsilon}_Y^*.\end{aligned} \qquad (5)$$

Accordingly, we denote such GWAS summary statistics of the exposure and the outcome as $\{\hat{\beta}_{Xi}^*, \hat{\beta}_{Yi}^*, \sigma_{Xi}^*, \sigma_{Yi}^*\}_{i=1}^{m}$.

### Transformation between marginal and conditional SNP-effect estimates

From Eq. (1), we have $\hat{\boldsymbol{\beta}}_X = (\mathbf{G}_X^T\mathbf{G}_X)^{-1}(\mathbf{G}_X^T\mathbf{X})$, where $\mathbf{G}_X$ is the $N_X \times m$ standardized genotype matrix and $\mathbf{X}$ is the standardized phenotype vector of length $N_X$. Then $\hat{\boldsymbol{\beta}}_X = \mathbf{R}_X^{-1}\hat{\boldsymbol{\beta}}_X^*$, where $\mathbf{R}_X$ is the LD matrix and $\hat{\boldsymbol{\beta}}_X^*$ is the GWAS estimate. Denote the (estimated) covariance matrix of the joint effect estimate ($\hat{\boldsymbol{\beta}}_X$) as $\boldsymbol{\Sigma}_X = \mathbf{R}_X^{-1}\boldsymbol{\Omega}_X\mathbf{R}_X^{-1}$, where $\boldsymbol{\Omega}_X = \mathbf{R}_X \cdot \boldsymbol{\sigma}_X^*\boldsymbol{\sigma}_X^{*T}$ is the covariance matrix of the marginal effect estimate $\hat{\boldsymbol{\beta}}_X^*$, " $\cdot$ " is the element-wise multiplication. In practice, if the individual genotype matrix for calculating $\mathbf{R}_X$ is not available, the LD matrix can be estimated using some publicly available reference panel denoted as $\mathbf{R}$, where $\mathbf{R} \approx \mathbf{R}_X$. Furthermore, if the GWAS estimates are not calculated on the standardized genotypes and phenotype, they can be approximated as $\hat{\beta}_{Xi}^* / \sqrt{\hat{\beta}_{Xi}^{*2} + (N_X - 2) \cdot \sigma_{Xi}^{*2}}$[49]. Similarly we can estimate $\hat{\boldsymbol{\beta}}_Y$ and $\boldsymbol{\Sigma}_Y$ from $\hat{\boldsymbol{\beta}}_Y^*$ and $\boldsymbol{\sigma}_Y^*$.

In the following sections, unless specified otherwise, we assume (asymptotic) normal distributions for the conditional-effect estimates $\hat{\boldsymbol{\beta}}_X \sim \mathcal{MVN}(\mathbf{b}_X, \boldsymbol{\Sigma}_X)$ and $\hat{\boldsymbol{\beta}}_Y \sim \mathcal{MVN}(\mathbf{b}_Y, \boldsymbol{\Sigma}_Y)$ with $\mathbf{b}_X = (b_{X1}, \ldots, b_{Xm})^T$ and $\mathbf{b}_Y = (b_{Y1}, \ldots, b_{Ym})^T$, which is reasonable given large sample sizes of GWAS data.

### MR-IVW and MR-egger

The inverse-variance weighted method (MR-IVW)[3] and MR-Egger regression[50] are two of the most widely used MR methods. These two methods are most often discussed in the context of independent SNPs/IVs, where MR-IVW and MR-Egger can be regarded as weighted linear regression (with weights equal to $\sigma_{Yi}^{*-2}$) of $\hat{\boldsymbol{\beta}}_Y^*$ on $\hat{\boldsymbol{\beta}}_X^*$, without and with the intercept term respectively. Both methods have previously been extended to account for correlated IVs and we will next give a brief overview of the existing methods, while details can be found in their corresponding references.

**Generalized IVW and egger.** To account for the correlations among IVs, MR-IVW and MR-Egger have been extended based on generalized weighted linear regression[14,15], and we refer to them as GIVW and GEgger throughout the paper. The GIVW and GEgger estimators are:

$$\hat{\theta}_{GIVW} = \left(\hat{\boldsymbol{\beta}}_X^{*T}\boldsymbol{\Omega}_Y^{-1}\hat{\boldsymbol{\beta}}_X^*\right)^{-1}\left(\hat{\boldsymbol{\beta}}_X^{*T}\boldsymbol{\Omega}_Y^{-1}\hat{\boldsymbol{\beta}}_Y^*\right), \qquad (6)$$

$$\begin{pmatrix}\hat{\alpha}_{GEgger}\\\hat{\theta}_{GEgger}\end{pmatrix} = \left[\left(\mathbf{1}, \hat{\boldsymbol{\beta}}_X^*\right)^T\boldsymbol{\Omega}_Y^{-1}\left(\mathbf{1}, \hat{\boldsymbol{\beta}}_X^*\right)\right]^{-1}\left[\left(\mathbf{1}, \hat{\boldsymbol{\beta}}_X^*\right)^T\boldsymbol{\Omega}_Y^{-1}\hat{\boldsymbol{\beta}}_Y^*\right], \qquad (7)$$

$$\hat{\theta}_{GEgger} = \frac{\left(\mathbf{1}^T\boldsymbol{\Omega}_Y^{-1}\mathbf{1}\right)\left(\hat{\boldsymbol{\beta}}_X^{*T}\boldsymbol{\Omega}_Y^{-1}\hat{\boldsymbol{\beta}}_Y^*\right) - \left(\mathbf{1}^T\boldsymbol{\Omega}_Y^{-1}\hat{\boldsymbol{\beta}}_X^*\right)\left(\mathbf{1}^T\boldsymbol{\Omega}_Y^{-1}\hat{\boldsymbol{\beta}}_Y^*\right)}{\left(\mathbf{1}^T\boldsymbol{\Omega}_Y^{-1}\mathbf{1}\right)\left(\hat{\boldsymbol{\beta}}_X^{*T}\boldsymbol{\Omega}_Y^{-1}\hat{\boldsymbol{\beta}}_X^*\right) - \left(\mathbf{1}^T\boldsymbol{\Omega}_Y^{-1}\hat{\boldsymbol{\beta}}_X^*\right)\left(\mathbf{1}^T\boldsymbol{\Omega}_Y^{-1}\hat{\boldsymbol{\beta}}_X^*\right)}, \qquad (8)$$

where $\boldsymbol{\Omega}_Y = \mathbf{R} \cdot \boldsymbol{\sigma}_Y^*\boldsymbol{\sigma}_Y^{*T}$ is the covariance matrix of $\hat{\boldsymbol{\beta}}_Y^*$. When the SNPs are independent, i.e., $\boldsymbol{\Omega}_Y$ becomes a diagonal matrix with the $i$th diagonal element $\sigma_{Yi}^{*2}$, GIVW and GEgger become the original MR-IVW and MR-Egger. The GIVW and GEgger methods are implemented in the R package `MendelianRandomization`[51].

## BOX 1
# cisMR-cML algorithm for a given $K$

**Require:** $\hat{\boldsymbol{\beta}}_\mathbf{X}, \hat{\boldsymbol{\beta}}_\mathbf{Y}, \boldsymbol{\Sigma}_\mathbf{X}, \boldsymbol{\Sigma}_\mathbf{Y}, \mathbf{K}, \theta^{(O)}, \mathbf{b}_\mathbf{X}^{(O)}$

  $\mathbf{t} \leftarrow \hat{\boldsymbol{\beta}}_\mathbf{Y} - \theta^{(O)}\mathbf{b}_\mathbf{X}^{(O)}$

  $\mathcal{A}_{\text{current}} \leftarrow \{\mathbf{i} : \sum_{j=1}^m \mathbf{l}(|\mathbf{t_i}| \leq |\mathbf{t_j}|) \leq \mathbf{K}\}$

  $\mathcal{A}_{\text{old}} \leftarrow \emptyset$

  $\mathbf{L}_{\text{old}} \leftarrow \mathbf{l}(\theta^{(O)}, \mathbf{b}_\mathbf{X}^{(O)}, \mathbf{t}^{\mathcal{A}_{\text{current}}})$

  **while** $\mathcal{A}_{\text{current}} \neq \mathcal{A}_{\text{old}}$ **do**

  $\mathcal{A}_{\text{old}} \leftarrow \mathcal{A}_{\text{current}}$

  $\{\tilde{\theta}, \tilde{\mathbf{b}}_\mathbf{X}, \tilde{\mathbf{r}}\} \leftarrow \arg\max_{\theta, \mathbf{b}_\mathbf{X}, \mathbf{r}} \mathbf{l}(\theta, \mathbf{b}_\mathbf{X}, \mathbf{r})$ under the constraint of $r_i \neq 0$ if $\mathbf{i} \in \mathcal{A}_{\text{current}}$

  $\mathbf{L}_{\text{current}} \leftarrow \mathbf{l}(\tilde{\theta}, \tilde{\mathbf{b}}_\mathbf{X}, \tilde{\mathbf{r}})$

  **if** $L_{\text{current}} > L_{\text{old}}$ **then**

  $L_{\text{old}} \leftarrow L_{\text{current}}$

  $\mathbf{t} \leftarrow (\hat{\boldsymbol{\beta}}_\mathbf{Y} - \tilde{\theta}\tilde{\mathbf{b}}_\mathbf{X})^2 / \mathbf{diag}(\boldsymbol{\Sigma}_\mathbf{Y})$

  $\mathcal{A}_{\text{current}} \leftarrow \{\mathbf{i} : \sum_{j=1}^m \mathbf{l}(\mathbf{t_i} \leq \mathbf{t_j}) \leq \mathbf{K}\}$

  $\{\hat{\theta}, \hat{\mathbf{b}}_\mathbf{X}, \hat{\mathbf{r}}\} \leftarrow \{\tilde{\theta}, \tilde{\mathbf{b}}_\mathbf{X}, \tilde{\mathbf{r}}\}$

  **end if**

  **end while**

  $\mathcal{I}_{\text{current}} \leftarrow (\mathcal{A}_{\text{current}})^\mathbf{c}$

  **return** $(\hat{\theta}, \hat{\mathbf{b}}_\mathbf{X}, \hat{\mathbf{r}}, \mathcal{A}_{\text{current}}, \mathcal{I}_{\text{current}})$

**LD-Aware (LDA) IVW and Egger.** LD-Aware (LDA) MR-IVW and MR-Egger are two other variants of MR-IVW and MR-Egger proposed by Barfield et al.[16] to account for LD among IVs, and we refer them to LIVW and LEgger throughout. These LDA-estimators are very similar to the generalized MR-IVW (GIVW) and MR-Egger (GEgger), except that the input data is the conditional estimates $\{\hat{\beta}_X^*, \hat{\beta}_Y^*\}$, instead of the marginal estimates $\{\hat{\beta}_X, \hat{\beta}_Y\}$:

$$\hat{\theta}_{LIVW} = \left(\hat{\boldsymbol{\beta}}_X^T \boldsymbol{\Sigma}_Y^{-1} \hat{\boldsymbol{\beta}}_X\right)^{-1} \left(\hat{\boldsymbol{\beta}}_X^T \boldsymbol{\Sigma}_Y^{-1} \hat{\boldsymbol{\beta}}_Y\right), \tag{9}$$

$$\hat{\theta}_{LEgger} = \frac{\left(\mathbf{1}^T \boldsymbol{\Sigma}_Y^{-1} \mathbf{1}\right)\left(\hat{\boldsymbol{\beta}}_X^T \boldsymbol{\Sigma}_Y^{-1} \hat{\boldsymbol{\beta}}_Y\right) - \left(\mathbf{1}^T \boldsymbol{\Sigma}_Y^{-1} \hat{\boldsymbol{\beta}}_X\right)\left(\mathbf{1}^T \boldsymbol{\Sigma}_Y^{-1} \hat{\boldsymbol{\beta}}_Y\right)}{\left(\mathbf{1}^T \boldsymbol{\Sigma}_Y^{-1} \mathbf{1}\right)\left(\hat{\boldsymbol{\beta}}_X^T \boldsymbol{\Sigma}_Y^{-1} \hat{\boldsymbol{\beta}}_X\right) - \left(\mathbf{1}^T \boldsymbol{\Sigma}_Y^{-1} \hat{\boldsymbol{\beta}}_X\right)\left(\mathbf{1}^T \boldsymbol{\Sigma}_Y^{-1} \hat{\boldsymbol{\beta}}_X\right)}, \tag{10}$$

with $\hat{\boldsymbol{\beta}}_X = \mathbf{R}^{-1}\hat{\boldsymbol{\beta}}_X^*$, $\hat{\boldsymbol{\beta}}_Y = \mathbf{R}^{-1}\hat{\boldsymbol{\beta}}_Y^*$ and $\boldsymbol{\Sigma}_Y = \mathbf{R}^{-1}\boldsymbol{\Omega}_Y \mathbf{R}^{-1}$. Comparing Eq. (6) with Eq. (9), and Eq. (8) with Eq. (10), we see that $\hat{\theta}_{LIVW} = \hat{\theta}_{GIVW}$, but in general $\hat{\theta}_{LEgger} \neq \hat{\theta}_{GEgger}$. The LDA-Egger method can be implemented with the code provided by the original authors (https://rbarfield.github.io/Barfield_website/pages/Rcode.html).

While the extensions of MR-IVW and MR-Egger allow for correlations among IVs, they inherit the same limitations in their corresponding original versions. For example, both MR-IVW and MR-Egger may yield biased causal inference unless all IVs are valid or under some stringent (so-called InSIDE) condition between the instrument strengths and their direct effects[15,17].

## New method: cisMR-cML

We propose a robust *cis*-MR method accounting for possible violations of any invalid IV assumptions, called cisMR-cML. Suppose we have the estimated joint/conditional associations of the $m$ SNPs with the exposure as $\hat{\boldsymbol{\beta}}_X = (\hat{\beta}_{X1}, \ldots, \hat{\beta}_{Xm})^T$ and its covariance matrix $\boldsymbol{\Sigma}_X$, and those with the outcome as $\hat{\boldsymbol{\beta}}_Y = (\hat{\beta}_{Y1}, \ldots, \hat{\beta}_{Ym})^T$ and $\boldsymbol{\Sigma}_Y$, which can be calculated from the GWAS summary statistics and LD matrix. The model for the proposed **cisMR-cML** is

$$\begin{aligned}\hat{\boldsymbol{\beta}}_X &\sim \mathcal{MVN}(\mathbf{b}_X, \boldsymbol{\Sigma}_X), \\ \hat{\boldsymbol{\beta}}_Y &\sim \mathcal{MVN}(\mathbf{b}_Y = \theta\mathbf{b}_X + \mathbf{r}, \boldsymbol{\Sigma}_Y),\end{aligned} \tag{11}$$

where $\boldsymbol{\Sigma}_X$ and $\boldsymbol{\Sigma}_Y$ are the covariance of $\hat{\boldsymbol{\beta}}_X$ and $\hat{\boldsymbol{\beta}}_Y$ respectively, $\theta$ is the causal effect of interest, $\mathbf{b}_X$ is a vector of the unknown joint effects of $m$ SNPs on the exposure, and $\mathbf{r}$ is a vector of the unknown direct effects on the outcome not mediated through the exposure. Note that $\mathbf{r}$ captures both the correlated and uncorrelated (horizontal) pleiotropic effects. Assuming the independence between the exposure GWAS dataset and the outcome GWAS dataset, we have the log-likelihood for the proposed model Eq. (11) (up to some constants):

$$\begin{aligned}l(\theta, \mathbf{b}_X, \mathbf{r}; \hat{\boldsymbol{\beta}}_X, \hat{\boldsymbol{\beta}}_Y, \boldsymbol{\Sigma}_X, \boldsymbol{\Sigma}_Y) = -\frac{1}{2}\Big[&\left(\hat{\boldsymbol{\beta}}_X - \mathbf{b}_X\right)^T \boldsymbol{\Sigma}_X^{-1}\left(\hat{\boldsymbol{\beta}}_X - \mathbf{b}_X\right) \\ &+ \left(\hat{\boldsymbol{\beta}}_Y - \theta\mathbf{b}_X - \mathbf{r}\right)^T \boldsymbol{\Sigma}_Y^{-1}\left(\hat{\boldsymbol{\beta}}_Y - \theta\mathbf{b}_X - \mathbf{r}\right)\Big].\end{aligned} \tag{12}$$

Under the constraint that the number of invalid IVs is $0 \leq K < m - 1$, we obtain the constrained maximum likelihood estimator (cMLE) by solving

$$\min_{\theta, \mathbf{b}_X, \mathbf{r}} - l(\theta, \mathbf{b}_X, \mathbf{r}; \hat{\boldsymbol{\beta}}_X, \hat{\boldsymbol{\beta}}_Y, \boldsymbol{\Sigma}_X, \boldsymbol{\Sigma}_Y) \text{ subject to } \sum_{i=1}^m I(r_i \neq 0) = K, \tag{13}$$

where $I(\cdot)$ is the indicator function, and $K$ is a tuning parameter representing the unknown number of invalid IVs to be determined by a model selection criterion to be discussed. The plurality valid condition implies $K < m - 1$.

For any candidate value of $K$, we use Box 1 to estimate the set of invalid IVs $\mathcal{A} := \{i : r_i \neq 0\}$ and the causal parameter $\theta$. Solving Eq. (13) is a best-subset selection problem, for which it is computationally too demanding to obtain a global solution for a moderate to large number of candidate IVs (i.e. $m$). We develop a heuristic algorithm based on iteratively ranking the variants based on their estimated pleiotropic effects. The algorithm is computationally efficient, but may not

converge to the global solution. As an alternative, we tried a recently proposed splicing algorithm that was proven to provide a global solution with a high probability[52]; it performed no better than the current algorithm in our simulations, though computationally more demanding. Alternatively, we also tried to use the Lasso penalty to shrink $r_i$ to zero for valid IVs (see Supplementary Section S2), which was also observed to have suboptimal performance in simulations (Supplementary Table S6). Hence we decided to use the heuristic and fast algorithm, which is to be shown to perform well in our numerical examples. Denote $\mathbf{a}^{\mathcal{A}}$ the vector whose $i$-th entry is $a_i$ if $i \in \mathcal{A}$, and is zero otherwise. The algorithm is given in Box 1.

As in the independent-IV case[7], it is notable that at the convergence the (estimated) invalid IVs (with $\hat{r}_i \neq 0$) do not contribute to estimating $\boldsymbol{\theta}$, and the resulting cMLE of $\boldsymbol{\theta}$ is the same as the maximum (profile) likelihood estimator being applied to all (selected) valid IVs. In practice, besides the default starting value of $\theta^{(0)} = 0, \mathbf{b}_X^{(0)} = \mathbf{0}$, we can use multiple random starts $\theta^{(0)}, \mathbf{b}_X^{(0)}$ and take the estimate with the largest likelihood among those from the multiple starting points as the cMLE under the constraint of $K$ invalid IVs.

Denote the estimates for a given $K$ as $\theta(K), \hat{\mathbf{b}}_X(K), \mathbf{r}(K), \hat{\mathcal{A}}(K), \hat{\mathcal{I}}(K)$. We select $K$ from a candidate set $\mathcal{K} \subseteq \{0, 1, \ldots, m - 2\}$ based on the following Bayesian information criterion (BIC):

$$\text{BIC}(K) = -2l(\hat{\theta}(K), \hat{\mathbf{b}}_X(K), \hat{\mathbf{r}}(K); \hat{\boldsymbol{\beta}}_X, \hat{\boldsymbol{\beta}}_Y, \boldsymbol{\Sigma}_X, \boldsymbol{\Sigma}_Y) + \log(N) \cdot K, \quad (14)$$

where $N = \min(N_X, N_Y)$. Then $\hat{K} = \arg\min_{K \in \mathcal{K}} \text{BIC}(K)$, $\hat{\mathcal{I}} = \hat{\mathcal{I}}(\hat{K})$, and the final causal estimate of Eq. (13) is $\hat{\theta} = \hat{\theta}(\hat{K})$. In the proposed algorithm, the resulting constrained maximum likelihood estimator is the same as the maximum profile likelihood estimator being applied to all IVs in $\hat{\mathcal{I}}$. The standard error of $\hat{\theta}$ can be estimated based on the observed Fisher information from the profile likelihood with IVs in $\hat{\mathcal{I}}$. With $\hat{\theta}$ and its corresponding standard error, the statistical inference is drawn based on the standard normal distribution, the theory of which is to be established in Theory.

The validity of the above inference relies on the selection consistency of invalid IVs (with $r_i \neq 0$), which may not always be realized with finite samples. Instead, to account for the uncertainty/variation in model selection, we will use data perturbation as before for better finite-sample statistical inference[7]. As shown in Lin et al.[53], the data perturbation procedure (on a GWAS summary dataset) is equivalent to bootstrapping the corresponding individual-level data. Briefly, for $b = 1, \ldots, B$, we generate perturbed conditional estimates $\hat{\boldsymbol{\beta}}_X^{(b)} \sim \mathcal{MVN}(\hat{\boldsymbol{\beta}}_X, \boldsymbol{\Sigma}_X)$ and $\hat{\boldsymbol{\beta}}_Y^{(b)} \sim \mathcal{MVN}(\hat{\boldsymbol{\beta}}_Y, \boldsymbol{\Sigma}_Y)$, and apply the estimation procedure above on the perturbed data to obtain $\hat{\theta}^{(b)}$. And we use the sample mean and sample standard deviation of $\hat{\theta}^{(1)}, \ldots, \hat{\theta}^{(B)}$ as the final causal estimate and its corresponding standard error.

## Modeling conditional effects versus marginal effects

A possible and seemingly effective alternative as in the current practice of MR analysis is to model marginal effects, instead of conditional effects, of SNPs (Eq. (11)). That is, we have

$$\begin{aligned} \hat{\boldsymbol{\beta}}_X^* &\sim \mathcal{MVN}(\mathbf{b}_X^* = \mathbf{R}\mathbf{b}_X, \mathbf{R} \cdot \boldsymbol{\sigma}_X^* \boldsymbol{\sigma}_X^{*T}), \\ \hat{\boldsymbol{\beta}}_Y^* &\sim \mathcal{MVN}(\mathbf{b}_Y^* = \mathbf{R}\mathbf{b}_Y = \theta\mathbf{b}_X^* + \mathbf{r}^*, \mathbf{R} \cdot \boldsymbol{\sigma}_Y^* \boldsymbol{\sigma}_Y^{*T}), \end{aligned} \quad (15)$$

where $\mathbf{r}^* = \mathbf{R}\mathbf{r}$. We can also have a similar relationship $b_{Yi}^* = \theta b_{Xi}^* + r_i^*$ as in Eq. (4). However, one pitfall is that, IVs without horizontal pleiotropy in the conditional model (i.e. with $r_i = 0$) may have $r_i^* \neq 0$ in the marginal model. For example, let $\mathbf{r} = (r_1, 0, \ldots, 0)^T$ and $r_1 \neq 0$, then $\mathbf{r}^* = \mathbf{R}\mathbf{r}$ will have non-zero elements for all $m$ SNPs when they are all correlated with the first SNP (i.e., the first column of $\mathbf{R}$ are all non-zeros). Hence, although the plurality condition holds in the conditional model, it is

violated in the marginal model. In general, the plurality condition is more likely to hold in the conditional model than in the marginal model. Therefore, in cisMR-cML, we use the joint/conditional effect estimates instead of the marginal effect estimates.

## Selection of genetic variants as IVs in cisMR-cML

In this section, we discuss which SNPs should be used in the proposed method and how to select them. First, it is crucial to include all $m$ SNPs associated with either the exposure or outcome, i.e. those in $\mathcal{I}_X \cup \mathcal{I}_Y$ in Fig. 1, not any of their proper subsets, and calculate their joint estimates with the exposure and the outcome. This is in striking contrast with the current practice of MR with independent IVs, where only SNPs significantly associated with the exposure (i.e., SNPs in $\mathcal{I}_X$) are used[7]. This is because, as shown in Fig. 1, conditional on $G_k$, $G_i$ in $\mathcal{I}_X \setminus \mathcal{I}_Y$ does not have a direct path to the outcome; but if we do not include SNPs in $\mathcal{I}_Y$ (e.g. $G_k$), then it will open alternative paths of all other correlated SNPs (with $G_k$) to the outcome not through the exposure. This will in turn break the plurality condition required by model identifiability since all SNPs will have direct effects on $Y$. On the other hand, such an issue is unlikely to occur when SNPs are all independent. We also note that, when we include SNPs in $\mathcal{I}_Y \setminus \mathcal{I}_X$, cisMR-cML is expected to select them out as invalid IVs.

In practice, to select these $m$ SNPs, we apply the COJO (Conditional and Joint association analysis) method[24] on the exposure and the outcome respectively to select SNPs in $\mathcal{I}_X$ and $\mathcal{I}_Y$. COJO is suitable in our application since it can identify SNPs that jointly are significantly associated with the phenotype via a stepwise selection procedure. It is applicable to both quantitative traits and case-control studies. Furthermore, it only uses GWAS summary statistics and an estimated LD matrix from a reference panel as in cisMR-cML.

## Theory

The proposed cisMR-cML enjoys nice asymptotic properties, including selection consistency of the proposed BIC and asymptotic normality of the cMLE. Here we state the assumptions and main conclusions with the proofs are relegated to the Supplementary.

**Assumption 1.** (Plurality valid condition.) Suppose that $\mathcal{A}_0 = \{i : r_i \neq 0\}$ is the index set of the true invalid IVs with a non-zero horizontal-pleiotropy effect, and $K_0 = |\mathcal{A}_0|$. For any $\mathcal{A} \subseteq \{1, \ldots, m\}$ and $|\mathcal{A}| = K_0$, if $\mathcal{A} \neq \mathcal{A}_0$, then there does not exist any constant $\tilde{\theta} \neq \theta$ such that $b_{Yi} = \tilde{\theta} b_{Xi}$ for all $i \in \mathcal{A}^C$.

**Assumption 2.** The joint effect estimates $\hat{\boldsymbol{\beta}}_X \sim \mathcal{MVN}(\mathbf{b}_X, \boldsymbol{\Sigma}_X)$ and $\hat{\boldsymbol{\beta}}_Y \sim \mathcal{MVN}(\mathbf{b}_Y, \boldsymbol{\Sigma}_Y)$ with the known covariance matrices $\boldsymbol{\Sigma}_X$ and $\boldsymbol{\Sigma}_Y$.

**Assumption 3.** (Orders of the variances and sample sizes.) Let $N = \min(N_X, N_Y)$, there exist positive constants $c_1, c_2$ such that $c_1/N \leq (\boldsymbol{\Sigma}_X)_{ij} \leq c_2/N$ and $c_1/N \leq (\boldsymbol{\Sigma}_Y)_{ij} \leq c_2/N$ for $i = 1, \ldots, m, j = 1, \ldots, m$, i.e., $\boldsymbol{\Sigma}_X$ and $\boldsymbol{\Sigma}_Y$ are $\Theta(1/N)$.

Assumption 1 is the plurality condition, which is equivalent to that in Theorem 1 of Guo et al.[48], a sufficient and necessary condition for the identifiability of model Eq. (4). Assumption 2 and 3 are reasonable given that GWAS summary data are usually based on large sample sizes. Then the following theorem gives the selection consistency and asymptotic normality and consistency of the proposed estimator.

**Theorem 1.** With Assumption 1 to 3 satisfied, if $K_0 \in \mathcal{K}$, we have $P(\hat{K} = K_0) \to 1$ and $P(\hat{\mathcal{A}}_{\hat{K}} = \mathcal{A}_0) \to 1$ as $N \to \infty$. And the constrained maximum likelihood estimator $\hat{\theta}$ of Eq. (13), combined with the use of the BIC selection criterion, is consistent for the true causal effect size $\theta_0$, and

$$\sqrt{V}(\hat{\theta} - \theta_0) \xrightarrow{d} \mathcal{N}(0, 1) \text{ as } N \to \infty,$$

where $V$ is the expected Fisher information for the profile log-likelihood with all IVs in $\mathcal{A}_0^C$ that can be consistently estimated by its sample version.

We note that, as implied by the constraint we use in Eq. (13), the invalid IVs in the proposed method are referred to as those in $\mathcal{I}_Y$ with a non-zero direct effect on the outcome ($r_i \neq 0$), which can be consistently selected out by the proposed BIC. On the other hand, although an irrelevant IV with $b_{Xi} = r_i = b_{Yi} = 0$ is also considered invalid, cisMR-cML will not select it out but including such an IV will not affect the validity of our inference as long as the conditions of Theorem 1 are satisfied. In summary, cisMR-cML is highly robust in the sense of allowing the presence of some invalid IVs violating any of the three valid IV assumptions; these invalid IVs can be more than half of all the IVs used.

## Simulations with autoregressive correlation structure

In this simulation study, we simulated the GWAS summary statistics largely following the simulation procedure used in the LDA-Egger paper[16]:

1. Generated the true joint effect of $|\mathcal{I}_X|$ SNPs on the exposure $b_{Xi} \sim \mathcal{N}(0,1)$, for $i \in \mathcal{I}_X$, and $b_{Xi} = 0$ for $i \notin \mathcal{I}_X$; rescaled the effects according to the proportion of variability in exposure due to SNPs: $\mathbf{b}_X = \sqrt{h_X^2/(\mathbf{b}_X^T \mathbf{R} \mathbf{b}_X)} \mathbf{b}_X$, where $h_X^2 = 0.05$, and $\mathbf{R}$ was the LD matrix generated from an autoregressive model with $\Sigma_{ij} = \rho^{|i-j|}$;

2. Generated the direct effects of $|\mathcal{I}_Y|$ SNPs on the outcome $r_i \sim \mathcal{N}(0,1)$ iid, with $K_1$ SNPs from $\mathcal{I}_X \cap \mathcal{I}_Y$ and $K_2$ SNPs from $\mathcal{I}_Y \setminus \mathcal{I}_X$; rescaled the direct effects according to the proportion of variability in outcome due directly to SNPs: $\mathbf{r} = \sqrt{h_Y^2/(\mathbf{r}^T \mathbf{R} \mathbf{r})} \mathbf{r}$, where $h_Y^2 = 0.05$;

3. Generated the true joint effects of SNPs on the outcome $\mathbf{b}_Y = \theta \mathbf{b}_X + \mathbf{r}$;

4. Generated the observed exposure GWAS estimates $\hat{\boldsymbol{\beta}}_X^* \sim \mathbf{R} \mathbf{b}_X + \mathbf{L}^T \boldsymbol{\epsilon}_X$, $\boldsymbol{\epsilon}_X \sim \mathcal{N}(\mathbf{0}, \frac{1-h_X^2}{N_X} \mathbf{I}_m)$, where $\mathbf{L}$ was the Cholesky decomposition of the LD matrix $\mathbf{R}$, and $N_X = 10000$. Note $\boldsymbol{\sigma}_X^* = \sqrt{\frac{1-h_X^2}{N_X}} \mathbf{1}_m$;

5. Generated the observed outcome GWAS estimates $\hat{\boldsymbol{\beta}}_Y^* \sim \mathbf{R} \mathbf{b}_Y + \mathbf{L}^T \boldsymbol{\epsilon}_Y$, $\boldsymbol{\epsilon}_Y \sim \mathcal{N}(\mathbf{0}, \frac{1-\theta^2 h_X^2 - h_Y^2}{N_Y} \mathbf{I}_m)$, and $N_Y = 50000$. Note $\boldsymbol{\sigma}_Y^* = \sqrt{\frac{1-\theta^2 h_X^2 - h_Y^2}{N_Y}} \mathbf{1}_m$.

In total $m = 10$ SNPs were generated. We considered two scenarios: (1) $|\mathcal{I}_X| = 10$; (2) $|\mathcal{I}_X| = 5$ and $K_2 = |\mathcal{I}_Y \setminus \mathcal{I}_X| = 2$. We note that in the second scenario, only 5 SNPs had effects on the exposure, 2 SNPs had effects on the outcome but not on the exposure, while 3 SNPs had no effect on either the exposure or the outcome. We would investigate the impact of only including the 5 SNPs in $|\mathcal{I}_X|$ in the analysis. In both scenarios, we varied $K_1$, the number of invalid IVs in $\mathcal{I}_X \cap \mathcal{I}_Y$.

Given the simulated GWAS summary statistics $(\hat{\boldsymbol{\beta}}_X^*, \boldsymbol{\sigma}_X^*, \hat{\boldsymbol{\beta}}_Y^*, \boldsymbol{\sigma}_Y^*)$, we transformed them to the conditional estimates as $\hat{\boldsymbol{\beta}}_X = \mathbf{R}^{-1} \hat{\boldsymbol{\beta}}_X^*$, $\hat{\boldsymbol{\beta}}_Y = \mathbf{R}^{-1} \hat{\boldsymbol{\beta}}_Y^*$ and $\boldsymbol{\Sigma}_X = \mathbf{R}^{-1}(\mathbf{R} \cdot \boldsymbol{\sigma}_X^* \boldsymbol{\sigma}_X^{*T}) \mathbf{R}^{-1}$, $\boldsymbol{\Sigma}_Y = \mathbf{R}^{-1}(\mathbf{R} \cdot \boldsymbol{\sigma}_Y^* \boldsymbol{\sigma}_Y^{*T}) \mathbf{R}^{-1}$. Note that in scenario (2), when we applied cisMR-cML and LEgger with the conditional estimates calculated only based on the 5 SNPs in $\mathcal{I}_X$, it was different from only using the corresponding 5 elements in $\hat{\boldsymbol{\beta}}_X$ and $\hat{\boldsymbol{\beta}}_Y$ calculated based on all 10 SNPs.

Each simulation setup was repeated 500 times. Throughout this simulation, cisMR-cML was implemented with 5 random starts where $\theta^{(0)} \sim \mathcal{U}(-0.5, 0.5)$ and $\mathbf{b}_X^{(0)} \sim \mathcal{N}(\hat{\boldsymbol{\beta}}_X, \boldsymbol{\Sigma}_X)$, and $B = 100$ data perturbations. IVW-IND, Egger-IND, cML-IND, GIVW and GEgger were implemented in the R package MendelianRandomization (v.0.9.0) with their default settings. LEgger was implemented in the R code provided at https://

rbarfield.github.io/Barfield_website/pages/Rcode.html. MR.LDP (https://github.com/QingCheng0218/MR.LDP), MR.Corr2 (https://github.com/QingCheng0218/MR.Corr2), MR.CUE (https://github.com/QingCheng0218/MR.CUE) and MRAID were implemented using corresponding R packages on GitHub, with parameters provided in their examples.

To investigate the method's robustness to weak IVs, we further considered the first scenario where $|\mathcal{I}_X| = 10$, $K_1 = 0$ and $\rho = 0.6$, but we reduced the instrument strength by reducing either the magnitude or the precision of IV-exposure association. To reduce the precision, we varied the sample size of the exposure GWAS $N_X \in \{500, 1000, 5000\}$. This corresponded to an average $F$-statistics (across 500 replications) of 3.6, 6.3 and 27.3 respectively. To reduce the magnitude of $b_{Xi}$, we varied $h_X^2 \in \{0.005, 0.01\}$, corresponding to an average $F$-statistics of 6.0 and 11.1 respectively.

As suggested by a reviewer, we also simulated IVs that were only weakly invalid in the sense that the pleiotropic effects on the outcome decreased at the same rate as sampling error $1/\sqrt{N_Y}$. Specifically, in the first scenario where $|\mathcal{I}_X| = 10$, $K_1 = 4$ and a moderate LD of $\rho = 0.6$, we directly generated the direct effects $r_i = \kappa/\sqrt{N_Y}$ in step 2 of the above simulation procedure. We varied the sample size of the outcome GWAS $N_Y \in \{5e4, 1e5, 5e5\}$ and $\kappa \in \{1, 5, 20\}$.

Finally, we considered an extreme scenario where outcome-associated SNPs ($\mathcal{I}_Y$) were uncorrelated or only weakly associated with exposure-associated SNPs ($\mathcal{I}_X$). Specifically, we simulated 10 SNPs with 5 SNPs in $\mathcal{I}_X$ and 5 SNPs in $\mathcal{I}_Y$ ($|\mathcal{I}_X \cap \mathcal{I}_Y| = 0$). SNPs within $\mathcal{I}_X$ and $\mathcal{I}_Y$ were correlated with an autoregressive structure of $\rho = 0.6$ respectively, while SNPs in $\mathcal{I}_X$ and SNPs in $\mathcal{I}_Y$ were uncorrelated ($\rho_{XY} = 0$) or weakly correlated ($\rho_{XY} = 0.1$). In other words, the LD matrix among SNPs in $\mathcal{I}_X \cup \mathcal{I}_Y$ was a block matrix, where the diagonal elements were two autoregressive LD blocks, and the off-diagonal elements were either all 0 or 0.1.

## Simulations with real LD patterns derived from UK Biobank data

To mimic a realistic application of the proposed method, we generated data directly from UK Biobank individual genotypes in this simulation study. We simulated data for the $j$-th individual as follows:

$$X_j = \sum_i b_{Xi} G_{ji} + U_j + \epsilon_{Xj},$$
$$Y_j = \theta X_j + \sum_i r_i G_{ji} + U_j + \epsilon_{Yj}, \tag{16}$$

where $U_j, \epsilon_{Xj}, \epsilon_{Yj} \sim \mathcal{N}(0,1)$ independently, $\theta \in \{0, 0.05\}$. We used the proteome-wide application on CAD as a reference to generate the data. Specifically, we considered the following two setups:

S1. 50 proteins with $|\mathcal{C}_X| \geq 5$ and $|\mathcal{C}_Y| \geq 1$ based on our real data analysis were randomly selected. For each protein, we specified the sets of SNPs $\mathcal{I}_X = \mathcal{C}_X$ and $\mathcal{I}_Y = \mathcal{C}_Y$ based on the COJO result in the real-data analysis, and the corresponding effect sizes $b_{Xi}$ and $r_i$ were set as the corresponding GWAS effect sizes in the real pQTL and CAD GWAS datasets respectively.

S2. 50 proteins were randomly selected. For each protein, we randomly selected 7 SNPs in the cis-region (with minor allele frequency (MAF) greater than 5%) in $\mathcal{I}_X$, and randomly selected 2 nearby SNPs (with MAF greater than 5%) in $\mathcal{I}_Y$, and ensured that the absolute pairwise correlations among SNPs in $\mathcal{I}_X \cup \mathcal{I}_Y$ were less than 0.95. The effects of SNPs in $\mathcal{I}_X$ on $X$ were generated from $b_{Xi} \sim \text{Unif}((-0.2, -0.1) \cup (0.1, 0.2))$ and the pleiotropic effects of SNPs in $\mathcal{I}_Y$ on $Y$ were generated from $r_i \sim \text{Unif}(0.1, 0.2)$.

After generating data for the exposure and the outcome, we randomly selected two non-overlapping sets of individuals, with sizes $N_X = 10000$ and $N_Y = 100000$, to perform GWAS on $X$ and $Y$ respectively. Note that the GWASs were performed for all SNPs in a specific cis-region using PLINK 2.0[54]. Additionally, we randomly selected a third

non-overlapping sample, with a size of $N_{ref} = 5000$, to mimic the use of an external reference panel for estimating LD structure.

Given the exposure and outcome GWAS datasets, we first applied GCTA-COJO (https://yanglab.westlake.edu.cn/software/gcta/#COJO, version 1.92.3beta3) with a significance threshold of $5 \times 10^{-6}$ to select SNPs jointly associated with $X$ or $Y$, denoted as $\mathcal{C}_X$ or $\mathcal{C}_Y$ respectively. We then applied the proposed cisMR-cML with SNPs in $\mathcal{C}_X \cup \mathcal{C}_Y$ with 5 random starts where $\theta^{(0)} \sim \mathcal{U}(-0.1, 0.1)$ and $\mathbf{b}_X^{(0)} \sim \mathcal{N}(\hat{\boldsymbol{\beta}}_X, \boldsymbol{\Sigma}_X)$, and $B = 100$ data perturbations. We also applied cisMR-cML-X and the current common practice of GIVW-X and GEgger-X with SNPs in $\mathcal{C}_X$ only. To apply MR.LDP, MR.Corr2, MR.CUE and MRAID, we first extracted SNPs that were marginally associated with $X$ with p-value $< 5 \times 10^{-6}$ and pruned the SNPs to ensure that any pairwise Pearson's absolute correlations were no more than 0.9. Finally, we considered the standard practice for polygenic MR by using only independent IVs associated with $X$. We performed LD clumping on the exposure GWAS dataset using the threshold of $r^2 = 0.001$, the default value used in `TwoSampleMR` package. Since the number of independent IVs was usually less than three, in which the independent version of MR-cML and Egger regression were not applicable, we only applied the IVW method, referred to as IVW-IND. Simulations were repeated 10 times per gene, with a total of 500 replicates.

### Reference panel used in real data applications

In the following two real data applications, we used the UK Biobank individual-level genotype data[55] as the reference panel. As the following analysis was based on GWAS datasets of (mostly) European ancestry, 337426 unrelated (field '22020'=1) and self-reported White-British individuals with similar genetic ancestry (field '22006'=1) in UK Biobank were used to calculate the LD matrix among SNPs.

### Drug-target MR application with the use of downstream biomarkers

GWAS summary data for both LDL cholesterol and testosterone were taken from the Neale Lab UK Biobank GWAS round 2 results (http://www.nealelab.is/uk-biobank/). And the GWAS summary data for CAD was obtained from CARDIoGRAMplusC4D Consortium[56]. We first extracted genetic variants located 500 kb on both sides of a gene, and retained those present in both the biomarker and CAD GWAS data, and confined our analysis to variants with missing genotypes $< 10\%$, minor allele frequency (MAF) $> 0.01$, Hardy-Weinberg equilibrium (HWE) $p > 1 \times 10^{-6}$ in the reference panel. Then we performed GCTA-COJO on the exposure (or outcome) GWAS data to select SNPs jointly associated with the exposure (or the outcome) at $p < 5 \times 10^{-6}$, denoted as $\mathcal{C}_X$ (or $\mathcal{C}_Y$) respectively.

We transformed the marginal association estimates $(\hat{\boldsymbol{\beta}}_X^*, \hat{\boldsymbol{\beta}}_Y^*)$ to the conditional estimates $(\hat{\boldsymbol{\beta}}_X, \hat{\boldsymbol{\beta}}_Y)$ for the SNPs in set $\mathcal{C}_X \cup \mathcal{C}_Y$, and calculated the corresponding covariance matrices $\boldsymbol{\Sigma}_X$ and $\boldsymbol{\Sigma}_Y$ according to Model. Then we applied cisMR-cML with $B = 100$ data perturbations with 5 random starts where $\theta^{(0)} \sim \mathcal{U}(-0.1, 0.1)$ and $\mathbf{b}_X^{(0)} \sim \mathcal{N}(\hat{\boldsymbol{\beta}}_X, \boldsymbol{\Sigma}_X)$, and LDA-Egger with the conditional estimates. We also applied GIVW-X and GEgger-X only using the marginal association estimates of SNPs in $\mathcal{C}_X$.

### A proteome-wide application to CAD

We confined our analysis to a list of 1034 proteins with $\geq 3$ identified pQTLs in the EA population according to Supplementary Table 6.1 in Zhang et al.[30]. For CAD GWAS, we used the one with a larger sample size of $N_Y = 547261$, which was a meta-analysis result of UK Biobank and CARDIoGRAMplusC4D[57].

The data preprocessing step was similar to that in Drug-target MR application with the use of downstream biomarkers, except that the LDL (or testosterone) GWAS data (i.e., exposure GWAS) was replaced by the pQTL dataset. We ran GCTA-COJO with $5 \times 10^{-6}$ as the p-value threshold on both the pQTL data and CAD GWAS data with the UK Biobank data as the reference panel to obtain $\mathcal{C}_X$ and $\mathcal{C}_Y$.

Here, we used a slightly less stringent p-value threshold of $5 \times 10^{-6}$, rather than the usual/default $5 \times 10^{-8}$, because omitting some relevant SNPs in $\mathcal{I}_X$ might lose power while omitting some SNPs in $\mathcal{I}_Y$ would affect the validity of the proposed method as discussed previously in Selection of genetic variants as IVs in cisMR-cML. We performed additional sensitivity analyzes using different p-value thresholds, which were detailed in Supplementary Section S5. We retained proteins with $\geq 3$ SNPs in $\mathcal{C}_X$, and excluded proteins with highly correlated (using "−cojo-collinear 0.9") SNPs in $\mathcal{C}_X$ and $\mathcal{C}_Y$. After the preprocessing step, 773 proteins remained to be analyzed next. We applied cisMR-cML and LEgger with the conditional estimates $(\hat{\boldsymbol{\beta}}_X, \hat{\boldsymbol{\beta}}_Y, \boldsymbol{\Sigma}_X, \boldsymbol{\Sigma}_Y)$ calculated on the SNP-set $\mathcal{C}_X \cup \mathcal{C}_Y$, as well as GIVW-X and GEgger-X on the pQTLs that were conditionally associated with the proteins (i.e., those in $\mathcal{C}_X$)[10,25]. We also applied the Wald-ratio test using the pQTL with the smallest marginal p-value for each protein. The number of SNPs used in cisMR-cML ranged from 3 to 20, with a mean of around 5; and running on an AMD 7763 processor, the computation times for cisMR-cML ranged from 1.6 to 680 seconds, with a mean of 20 seconds.

We further conducted colocalization analysis on the significant proteins with an FDR-adjusted p-value less than 0.05 (using "p.adjust(method='fdr')" in R). Colocalization analysis has been more regularly used and strongly recommended in practice following MR analysis[36]. In this analysis, we used a Bayesian colocalization method called COLOC[58], where a high H4-PP suggested the protein and CAD shared the same causal variant at the locus, while a high H3-PP suggested the protein and CAD had different causal variants at the locus. The former case supported the significant result from MR, however, the latter case suggested the significant MR result may be driven by genetic confounding through LD between pQTLs and CAD-associated SNPs, e.g. SNPs in $\mathcal{I}_Y \setminus \mathcal{I}_X$. COLOC was implemented with `coloc.abf()` in the R package `coloc` (v.5.2.3) with the default setting.

### Reporting summary

Further information on research design is available in the Nature Portfolio Reporting Summary linked to this article.

## Data availability

The GWAS summary datasets used in the real data analysis are all publicly available at the URLs below. ARIC pQTL, http://nilanjanchatterjeelab.org/pwas/; GWAS Catalog studies GCST003116 and GCST005194 for coronary artery disease, https://www.ebi.ac.uk/gwas/home; Neale lab UK Biobank round 2 GWAS of LDL and testosterone, https://www.nealelab.is/uk-biobank/. The UK Biobank individual-level data are available under restricted access. Researchers can apply for access at https://www.ukbiobank.ac.uk/. Access to UK Biobank individual-level data was approved through UKB Application #35107. The processed pQTL data used in the real data application are available at https://doi.org/10.6084/m9.figshare.25411957. Source data are provided with this paper.

## Code availability

R code for simulation studies and real data analysis is available at https://github.com/ZhaotongL/cisMR-paper[59]. The software for cisMR-cML is publicly available on GitHub at https://github.com/ZhaotongL/cisMRcML[60].

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

## Acknowledgements

This research was supported by NIH grants R01 AG065636 (to Z.L. and W.P.), R01 AG069895 (W.P.), RF1 AG067924 (W.P.), U01 AG073079 (W.P.), R01 AG074858 (W.P.), R01 HL116720 (W.P.) and R01 GM126002 (W.P.), and by the Minnesota Supercomputing Institute at the University of Minnesota (to Z.L. and W.P.). This manuscript is largely based on a chapter of the first author's PhD dissertation at the University of Minnesota. There is no issue of copyright with the dissertation; see https://pq-static-content.proquest.com/collateral/media2/documents/umi_embargorest.pdf.

## Author contributions

Z.L. and W.P. conceived the methods and wrote the manuscript. Z.L. performed all the data analysis and simulation studies.

## Competing interests

The authors declare no competing interests.
