## [Peer Review File · Nature Communications]

A robust cis-Mendelian randomization method with application to drug target discoveryREVIEWER COMMENTS

Reviewer #1 (Remarks to the Author):

The paper presents a novel method for pleiotropy-robust Mendelian randomization (MR) with correlated instruments, by extending the MR-cML method previously developed by one of the authors for MR with independent instruments. The new method is assessed in a simulation study and then implemented in two real-data cis-MR applications and a proteome-wide analysis of proteins associated with coronary artery disease (CAD) risk.

Although several pleiotropy-robust MR methods have been developed over the recent years, the literature on methods with correlated instruments is lacking. From that perspective, I believe the authors' work is both novel and potentially impactful. I enjoyed reading the paper and have only a few minor comments before accepting it for publication:

- 1) Cis-MR methods often become numerically unstable when SNPs are highly correlated (Burgess et al. 2017), as they need to work with and possibly invert an ill-conditioned genetic correlation matrix. Did the authors encounter any such issues with their cisMR-cML method?
- 2) The proposed method optimizes the summary-data likelihood (10) repeatedly, each time imposing the constraint that exactly K of the m SNPs are invalid and varying K between 0 and $m-1$. Could an alternative be to use penalization (e.g. by lasso) and optimize the likelihood once, letting the coefficients r_i shrink to zero for valid SNPs?
- 3) Implementing the cisMR-cML method seems a computationally demanding task, especially given the need to use bootstrap to compute standard errors. How long did the method take to run e.g. in the real-data applications?
- 4) The proposed method was slightly conservative in the simulation results presented in the paper, its empirical Type I error being consistently below nominal levels and ranging from 1% to 4.2% for a null causal effect. Should this be taken as a feature of the method in general, and if so, what is causing it?
- 5) An interesting question in practice is how common pleiotropy is in cis-MR analyses. Intuitively, it could be argued that SNPs in the same gene region may act through the same biological mechanism and hence be less likely to exhibit heterogeneous effects than SNPs across different regions in a genome-wide MR analysis. The cisMR-cML method can be used to identify SNPs that exhibit pleiotropic effects ($r_i \neq 0$). Did the authors detect such pleiotropic SNPs in the PCSK9 and SHBG regions in their applications? And how common was pleiotropy in their proteome-wide analysis?
- 6) There is a slight inconsistency in the PCSK9 application, where the authors claim that GCTA-COJO selected 9 SNPs associated with LDL cholesterol in the region but then mention that $|I_X|=8$. Please clarify.

Reviewer #3 (Remarks to the Author):

The authors propose a robust method for estimation in cis-Mendelian randomization studies. The method (cisMR-cML) extends a previously proposed approach (MR-cML; Xue et al., 2021, AJHG) to allow the use of correlated SNPs as instrumental variables (IVs). The authors suggest there are two main differences from the MR-cML method. The first is that the joint (or conditional) effects of correlated SNPs are modeled rather than their marginal effects – this approach has also been taken in several existing proposals for MR analysis with correlated instruments. The second is that SNPs that are associated with the outcome are also considered as candidate IVs, because otherwise the model may not be able to distinguish the effects of valid IVs from the effects of invalid IVs that are in correlation with the valid IVs – this proposal is new because most MR studies select SNPs according to their relevance for the exposure only. The method is applied to real data examples that study drug targets for coronary artery disease.

The paper is well written. However, the recommendation that SNPs associated with the outcome should be considered as candidate instruments needs further clarification. Relatedly, the quite

limited simulation study makes it difficult to gauge the relative merits and limitations of the proposed approach for typical cis-MR studies. It would seem there are several practical trade-offs that would be important to explore, but they are not currently discussed in the paper. I also have some statistical concerns regarding the validity of the proposed approach. My specific comments are listed below.

1. Plurality condition for cis-MR. The suitability of the approach for cis-MR studies beyond just using correlated SNPs should be discussed. As the authors note on p.4 lns.57–58, there have been several recently proposed methods that account for both LD and horizontal pleiotropy, but they rely on different modeling assumptions on the unobservable pleiotropic effects. Is the plurality condition more or less plausible in typical cis-MR studies compared with more general MR applications where candidate IVs can be uncorrelated? If the candidate IVs are quite correlated, the plurality condition would seem to be a bit “all or nothing” – i.e. that in typical studies either all instruments are valid or they are all invalid.

2. Highly correlated SNPs. Relatedly to point 1, the application of the method in practice seems to be heavily tied to GCTA-COJO to find conditionally relevant candidate IVs. When taking the union of the sets of conditionally relevant SNPs for the exposure and the outcome, is there a risk that some of the candidate IVs are highly correlated? E.g. at correlation greater than 0.8? In general, what was the maximum correlation between SNPs selected using GCTA-COJO in the data examples? Is the method quite robust to using highly correlated candidate IVs?

3. Outcome-associated SNPs. The authors are correct to highlight that omitting SNPs that are associated with the outcome could bias the evidence from valid IVs in a linear model of joint SNP effects (if the valid IVs are correlated with the invalid IVs). The recommendation by the authors is then that SNPs that are associated with the outcome should be included. To me, this is not an obvious conclusion. This choice may involve considerable trade-offs that should be investigated:

(a) on whether the plurality condition is more or less likely to hold: the authors suggest the plurality condition is less likely to hold if SNPs associated with the outcome are not considered as candidate IVs (p.23, lns.444–449). If there are a set of candidate SNPs only relevant for the exposure (lets call them C1) that are uncorrelated with many invalid IVs only relevant for the outcome (lets call them C2), could the inclusion of these many invalid IVs sway the balance of valid versus invalid candidate IVs? Even if the set of SNPs in C1 were weakly associated with SNPs in C2, would the method in practice give more biased estimates when considering the union of C1 and C2 as candidate IVs, rather than just C1?

(b) on a “winner's curse” bias: if candidate IVs are selected based on their measured associations with the outcome, would this compromise the ability to test a null hypothesis of a zero causal effect? There may be an issue in terms of a winner's curse-type bias. That is, due to sampling uncertainty in the measured genetic associations, retaining SNPs based on how strongly they are associated with the outcome may bias the evidence towards concluding a non-zero causal effect. The current simulation setting with 10 SNPs may not reflect this, but under a more realistic setup that selects from a large number of correlated SNPs based on their p-values (as in the real data example), I wonder if this trade-off from including outcome-associated SNPs may be more apparent.

(c) on the number of SNPs used to fit a joint model: including outcome-associated SNPs would increase the regressors in the joint SNP model (Equations (1) and (2) in Section 4.1). Is there a concern that, with a large number of correlated candidate IVs, the co-variance matrices (Σ_X and Σ_Y) in Equation (9) may be more difficult to estimate? Would the p-value threshold of 5×10^{-6} for GCTA-COJO that is recommended by the authors (p.13 lns.263–264) be expected to lead to a manageable number of SNPs to fit this joint model?

4. LD misspecification. As the authors note (p.6 lns.102–103) the LD matrix is often estimated from a publicly available reference panel, and two-sample studies are vulnerable to misspecified or mismeasured LD estimates. An extended simulation study should explore sensitivity of the proposed method to misspecification/mismeasurement in LD estimates. Also, is an autoregressive LD structure used in simulations realistic for typical cis-MR studies? If GCTA-COJO is the preferred choice to select candidate IVs, then are there concerns regarding the stability of the results with poor LD estimates compared with a clumping approach that offers some control over how correlated candidate IVs are permitted to be?

5. Slightly invalid IVs. Does the proposed method rely on a clear separation between what a valid and what an invalid IV is? In practice, this separation may not always be clear, and some SNPs may be slightly invalid IVs. This is usually modeled by local misspecification, where the direct IV effects on the outcome are “local to zero” (decreasing at the same rate as sampling errors $1/\sqrt{n}$). In this case where some IVs are slightly invalid, would the proposed method based on the plurality rule select all slightly invalid IVs? Would this harm inferences (confidence intervals/hypothesis tests)?

6. Power gain through correlated IVs. The premise of the proposed method is to use correlated IVs to improve power. e.g. p.3 ln.47 discusses the “possibly severe loss of power due to only one or few independent SNPs remaining”. Has this been demonstrated in simulation or by existing work? i.e. given that many SNPs in a single gene region are likely to be highly correlated, is there a significant increase in power from using correlated IVs?

7. Weak IVs. How does the proposed method perform if only weak IVs are available? On p.1 ln.12, the proposed method is described as “robust to violation of any of the three valid IV assumptions”. To me, this would imply that the method offers robustness to a weak IV problem where no strong IVs are available. Is the approach robust to weak IVs or are they selected out?

8. Type I error rates. For the simulation results, I was not able to find the nominated significance level. e.g. are these the results corresponding to a 5% level test? If so, the type I error rates of cisMR-cML under the null hypothesis of no causal effect appear to be quite a bit lower than the nominal level of 5% in Tables 1 and 2. While an under-rejection issue under the null hypothesis is less of a problem compared to larger type I error rates, if the rejection rates under the null hypothesis are much lower than the nominated significance level, this may suggest underlying problems with the theoretical results or code. Could the authors please clarify what the nominated significance level is. If it is 5%, some clarification on this potential under-rejection issue would be useful.

9. Comparator methods. The comparator methods considered in simulation are quite limited, making it difficult to understand whether the advantages of the cisMR-cML method are due to robustness to many invalid IVs, conditional/joint modeling of SNP effects, or because methods such as IVW are generally less powerful and less robust than likelihood-based approaches. The authors cite a number of recently developed methods on p.4 ln.57–58 that are not considered as comparator methods in simulation. Some of them, for example, MR-Corr2, also model the joint effects of SNPs. While an exhaustive simulation study of all available methods is not feasible nor the objective, the authors could also consider in simulation a few more state-of-the-art approaches to allow a clearer illustration of the relative merits and drawbacks of the proposed approach.

10. Other (minor) comments.

(a) p.3 ln.42, please check grammar on “(as potential drug target)”.

(b) p.4 ln.50, “only few cis-MR methods are robust to the violation of the IV assumptions.” seems at odds with the sentence in the same paragraph “There are several recently proposed methods to account for both LD and horizontal pleiotropy, such as MR-LDP, MR-Corr, MR-CUE, MRAID and RBMR”.

Authors' one-to-one responses

We appreciate the insightful, constructive and helpful comments from the two reviewers. Accordingly, we have made concerted efforts to carefully address each question raised as detailed below. The new or modified parts are highlighted in blue in the main text and Supplementary.

Editorial requests

“In particular, please address all of reviewer 3’s comments as thoroughly as possible, please make clearer the novelty of your method and please benchmark against previous methods to showcase the advance your method provides.”

Reply: We would like to thank you and the reviewers for your time and effort in reviewing our manuscript. We have addressed and incorporated all of Reviewer 3’s comments in the revised manuscript. We have now rephrased and highlighted the novelty of the proposed method in Section 2.1 and added previous methods in benchmarking.

First, compared to the previous method MR-cML, which requires the use of *independent* IVs in the analysis, cisMR-cML allows for correlated IVs and, therefore is more suitable for cis-MR applications such as drug target discovery. As detailed in our response to Reviewer 3 Point 6, using only independent IVs has three potential issues. The first issue is the potential reduction of power, and the second issue is the failure to implement MR-cML. Per your suggestion, we have benchmarked against MR-cML (denoted as cML-IND) in the simulation studies. As shown in Figure 2A, when $\rho = 0.2$ (the first column), cML-IND had lower power and larger RMSE than cisMR-cML; and when $\rho = 0.6$ or 0.8 , we failed to apply cML-IND (MR-cML) as only 2 or 1 independent IVs present in the region. This phenomenon was also observed in the second set of simulation studies based on our real-data analysis. The final issue is that using independent SNPs may introduce pleiotropic effects through LD. As shown in Figure 2B, cML-IND had highly inflated type-I error, due to the correlation of the independent IVs used in the analysis and some omitted invalid IVs.

As noted by Reviewer 3, our proposed cisMR-cML has two modeling novelties distinct

from MR-cML, in addition to accommodating correlated IVs: (1) it models the joint effects of correlated SNPs instead of their GWAS marginal effects; (2) it incorporates outcome-associated SNPs alongside the conventional practice of utilizing exposure-associated SNPs. These two points are discussed in Sections 4.5 and 4.6. In the original manuscript, the implementation of cisMR-cML-X was shown to have inflated type-I error in scenario 2 of the first set of simulations (now Figure 3), which highlighted the severe consequence of ignoring outcome-associated SNPs.

To strengthen the importance of point (2), we have further added several simulation experiments following Reviewer 3 Point 3. We again observed better performance of cisMR-cML than cisMR-cML-X. To highlight the importance of point (1), we have now added the implementation of cisMR-cML-Marg, which modeled the marginal genetic effects as shown in Eq. (12). As detailed in Supplementary Section S3.2, cisMR-cML-Marg had highly inflated type-I errors in the presence of 4 invalid IVs in scenario 1 of the first set of simulations. This occurred because, when the pleiotropic effects of the invalid IVs were absorbed in the marginal genetic effects, all IVs became invalid and the marginal model imposed by cisMR-cML-Marg was not identifiable. This new implementation in benchmarking highlights the importance of our point (1).

1 Reviewer 1

Introductory comment: *“The paper presents a novel method for pleiotropy-robust Mendelian randomization (MR) with correlated instruments, by extending the MR-cML method previously developed by one of the authors for MR with independent instruments. The new method is assessed in a simulation study and then implemented in two real-data cis-MR applications and a proteome-wide analysis of proteins associated with coronary artery disease (CAD) risk.*

Although several pleiotropy-robust MR methods have been developed over the recent years, the literature on methods with correlated instruments is lacking. From that perspective, I believe the authors’ work is both novel and potentially impactful. I enjoyed

reading the paper and have only a few minor comments before accepting it for publication.”

Reply: We would like to thank the reviewer for his/her encouraging comment!

Major comments:

1. *“Cis-MR methods often become numerically unstable when SNPs are highly correlated (Burgess et al. 2017), as they need to work with and possibly invert an ill-conditioned genetic correlation matrix. Did the authors encounter any such issues with their cisMR-cML method?”*

Reply: Thank you for the comment! No, we didn't encounter such issues in the implementation of cisMR-cML for two reasons. First, we propose to apply conditional analysis COJO to select IVs that are *jointly* associated with the exposure (SNPs in I_X) or the outcome (SNPs in I_Y), which usually leaves us a relatively small number of IVs and the LD matrix among them is manageable. And in the implementation of COJO, the collinearity problem is prevented using the default value of `--cojo-collinear 0.9`. Second, in the real data analysis, we had a further step to exclude proteins with SNPs in I_X and I_Y that were highly correlated (now Line 718). As a result, the SNPs used in the final analysis in cisMR-cML will not have collinearity or ill-conditioned genetic correlation matrix issues. Nonetheless, we have now added an initial checking step in our R package to warn against ill-conditioned correlation matrix.

2. *“The proposed method optimizes the summary-data likelihood (10) repeatedly, each time imposing the constraint that exactly K of the m SNPs are invalid and varying K between 0 and $m-1$. Could an alternative be to use penalization (e.g. by lasso) and optimize the likelihood once, letting the coefficients r_i shrink to zero for valid SNPs?”*

Reply: Thank you for the excellent suggestion! Per your suggestion, we have investigated the Lasso implementation of the proposed log-likelihood Eq. (10) with a lasso

penalty on the pleiotropic effects (referred to as cisMR-Lasso):

$$\arg \min_{\theta, \mathbf{b}_X, \mathbf{r}} (\hat{\boldsymbol{\beta}}_X - \mathbf{b}_X)^T \boldsymbol{\Sigma}_X^{-1} (\hat{\boldsymbol{\beta}}_X - \mathbf{b}_X) + (\hat{\boldsymbol{\beta}}_Y - \theta \mathbf{b}_X - \mathbf{r})^T \boldsymbol{\Sigma}_Y^{-1} (\hat{\boldsymbol{\beta}}_Y - \theta \mathbf{b}_X - \mathbf{r}) + \lambda \sum_{i=1}^m |r_i|.$$

First, we would like to point out that, in this penalized log-likelihood problem, we do not only optimize the objective function once. This is because we still need to tune the tuning parameter λ , which controls the sparsity of invalid IVs. Following the independent-IV version of MR-Lasso⁴, we search λ from (0.1, 0.2, ..., 4.9, 5.0, 5.2, 5.4, ..., 9.8, 10.0) using the same BIC criteria as in cisMR-cML by default. And in fact, in the context of cis-MR application where the number of IVs used in the proposed cisMR-cML is relatively small, this takes a much longer time than searching K from $0 \leq K < m - 1$. Furthermore, as discussed in Xue et al.⁵, the effectiveness of MR-Lasso depends on a specified candidate set of λ , which may be difficult to specify a priori. As shown in the additional simulation results, cisMR-Lasso had suboptimal performance compared to cisMR-cML. Specifically, in the presence of invalid IVs (Supplementary Table S6), cisMR-Lasso had slightly inflated type-I error and biased estimates; and in terms of the selection of invalid IVs, cisMR-Lasso-BIC also had a lower true positive rate (selecting the true invalid IVs) and a higher false positive rate (identifying the true valid IVs as invalid) than cisMR-cML-BIC.

We have detailed the Lasso implementation in Supplementary Section S2, and its performance in the simulation studies is provided in Supplementary Section S3.3.

3. *“Implementing the cisMR-cML method seems a computationally demanding task, especially given the need to use bootstrap to compute standard errors. How long did the method take to run e.g. in the real-data applications?”*

Reply: Thank you for the comment! The computation times for the cisMR-cML method (excluding the time to run GCTA-COJO) in our real-data application ranged from 1.6 to 680 seconds, with a median of 9 seconds, and a mean of 20 seconds. As you correctly pointed out, the computation bottleneck is in the bootstrap step (and the search for the number of invalid IVs K), which can be fully paralleled to reduce

computation time if necessary. In the current application, the time reported was without parallelization and a simple for-loop in R was used.

We have now reported the computation time in Lines 723-726.

4. *“The proposed method was slightly conservative in the simulation results presented in the paper, its empirical Type I error being consistently below nominal levels and ranging from 1% to 4.2% for a null causal effect. Should this be taken as a feature of the method in general, and if so, what is causing it? ”*

Reply: Thanks for pointing this out. Yes, we agree that cisMR-cML was conservative, especially in the absence of invalid IV. We would like to clarify that, the current (and default) implementation of cisMR-cML uses the data perturbation scheme. Simply using the BIC selection (without data perturbation) may give us inflated type-I error in the presence of invalid IVs for finite sample size problems, and the data perturbation approach is proposed to account for the uncertainty in model selection (i.e. the selection of invalid IVs). In simulation studies, we found a larger estimated standard error (average across 500 replicates) than the standard deviation of causal estimates (across 500 replicates) for cisMR-cML, resulting in a conservative type-I error rate. This conservative feature of the data perturbation implementation has also been observed before in the independent-IV version of MR-cML⁵ and MVMR-cML³.

We have now acknowledged this conservative feature as a potential limitation in the Discussion section (Lines 400-403), which will be investigated further in future studies.

5. *“An interesting question in practice is how common pleiotropy is in cis-MR analyses. Intuitively, it could be argued that SNPs in the same gene region may act through the same biological mechanism and hence be less likely to exhibit heterogeneous effects than SNPs across different regions in a genome-wide MR analysis. The cisMR-cML method can be used to identify SNPs that exhibit pleiotropic effects ($r_i \neq 0$). Did the authors detect such pleiotropic SNPs in the PCSK9 and SHBG regions in their applications? And how common was pleiotropy in their proteome-wide analysis?”*

Reply: Thank you for the great question! From a biological standpoint, SNPs in the same gene region may still have different biological mechanisms. For example, transcription factors (TFs) play an important role in mediating gene-regulatory mechanism in many human diseases. Genetic variation in TF-binding sites will potentially influence the binding affinity or efficiency of the TF, which may subsequently affect the production of the associated RNA and proteins. As most TF-binding sites are small, a typical human gene (>20kb) will likely contain multiple potential binding sites for most TFs², suggesting even SNPs located in the same gene region may have different biological mechanisms.

In terms of the real data application, we would like to first clarify two different sets of invalid IVs $\mathcal{I}_Y = \{i : r_i \neq 0\}$. One is those in $\mathcal{I}_Y \setminus \mathcal{I}_X = \{i : r_i \neq 0, b_{Xi} = 0\}$, i.e., SNPs jointly associated with the outcome but not the exposure; and the other is those in $\mathcal{I}_Y \cap \mathcal{I}_X = \{i : r_i \neq 0, b_{Xi} \neq 0\}$, i.e., SNPs associated with the exposure and the outcome. We used COJO in the real data application to select SNPs in \mathcal{I}_X and \mathcal{I}_Y , referring to the selected sets as \mathcal{C}_X and \mathcal{C}_Y respectively.

In the first application, COJO selected $|\mathcal{C}_Y \setminus \mathcal{C}_X| = 1$ in the PCSK9 region. While cisMR-cML-BIC didn't detect this SNP as invalid, cisMR-cML detected this as invalid over 10% of the time among the 100 data perturbations. No invalid IV in $\mathcal{C}_Y \cap \mathcal{C}_X$ was detected. For the testosterone SHBG region, COJO didn't select any SNPs associated with CAD (i.e., $|\mathcal{C}_Y| = 0$), and no invalid IV was detected. In the second application, among the 773 proteins, 183 had $|\mathcal{C}_Y \setminus \mathcal{C}_X| \geq 1$, and cisMR-cML-BIC detected over 98.7% of such invalid IVs. Furthermore, 30 protein-CAD pairs had one invalid IV in $\mathcal{C}_Y \cap \mathcal{C}_X$ detected by cisMR-cML-BIC. It is also noted that in the finite sample size problem, the selection of invalid IVs based on BIC may not be perfect, especially in the presence of weak invalid IVs. With data perturbation, cisMR-cML has detected 310 protein-CAD pairs had at least one invalid IV in $\mathcal{C}_Y \cap \mathcal{C}_X$ over 10% of the time among the 100 data perturbations.

6. *“There is a slight inconsistency in the PCSK9 application, where the authors claim*

that GCTA-COJO selected 9 SNPs associated with LDL cholesterol in the region but then mention that $|I_X| = 8$. Please clarify."

Reply: We apologize for the confusion. We meant to say that COJO selected 9 SNPs located in the *PCSK9* region, 8 of which were associated with LDL and one was associated with CAD. We have corrected the sentence now.

2 Reviewer 3

Introductory comment:

"The authors propose a robust method for estimation in cis-Mendelian randomization studies. The method (cisMR-cML) extends a previously proposed approach (MR-cML; Xue et al., 2021, AJHG) to allow the use of correlated SNPs as instrumental variables (IVs). The authors suggest there are two main differences from the MR-cML method. The first is that the joint (or conditional) effects of correlated SNPs are modeled rather than their marginal effects – this approach has also been taken in several existing proposals for MR analysis with correlated instruments. The second is that SNPs that are associated with the outcome are also considered as candidate IVs, because otherwise the model may not be able to distinguish the effects of valid IVs from the effects of invalid IVs that are in correlation with the valid IVs – this proposal is new because most MR studies select SNPs according to their relevance for the exposure only. The method is applied to real data examples that study drug targets for coronary artery disease.

The paper is well written. However, the recommendation that SNPs associated with the outcome should be considered as candidate instruments needs further clarification. Relatedly, the quite limited simulation study makes it difficult to gauge the relative merits and limitations of the proposed approach for typical cis-MR studies. It would seem there are several practical trade-offs that would be important to explore, but they are not currently discussed in the paper. I also have some statistical concerns regarding the validity of the proposed approach. My specific comments are listed below."

Reply: We are very grateful for your helpful summary, insightful comments, and valuable

suggestions, which have substantially enhanced the manuscript. We have made every effort to address the comments you raised in the following point-by-point responses.

1. *“Plurality condition for cis-MR. The suitability of the approach for cis-MR studies beyond just using correlated SNPs should be discussed. As the authors note on p.4 lns.57–58, there have been several recently proposed methods that account for both LD and horizontal pleiotropy, but they rely on different modeling assumptions on the unobservable pleiotropic effects. Is the plurality condition more or less plausible in typical cis-MR studies compared with more general MR applications where candidate IVs can be uncorrelated? If the candidate IVs are quite correlated, the plurality condition would seem to be a bit “all or nothing” – i.e. that in typical studies either all instruments are valid or they are all invalid.”*

Reply: Thank you for the great question! You are absolutely right that in the context of cis-MR analysis where IVs are quite correlated, the plurality condition is likely to be ‘all or nothing’ (if no special action is taken). And we totally agree with you that the plurality assumption is more likely to be violated in cis-MR analysis if we do not handle correlated SNPs properly. In fact, this is exactly a main motivation for this work to model the conditional genetic effects instead of marginal genetic effects, and to include the SNPs affecting the outcome in the analysis. These two proposals both try to alleviate the breakdown of the plurality condition in cis-MR analysis, which has been discussed in Sections 4.5 and 4.6.

Furthermore, even though we have properly adjusted for potential pleiotropic effect introduced by LD correlation, it is still likely that some SNPs exhibit biological pleiotropy (i.e. invalid IVs in $\mathcal{I}_X \cap \mathcal{I}_Y$). For example, transcription factors (TFs) play an important role in mediating gene-regulatory mechanism in many human diseases. Genetic variation in TF-binding sites will potentially influence the binding affinity or efficiency of the TF, which may subsequently affect the production of the associated RNA and proteins. As most TF-binding sites are small, a typical human gene (>20kb) will likely contain multiple potential binding sites for most TFs², suggesting even the SNPs lo-

cated in the same gene region may have different biological mechanisms to influence the exposure/outcome.

2. *“Highly correlated SNPs. Relatedly to point 1, the application of the method in practice seems to be heavily tied to GCTA-COJO to find conditionally relevant candidate IVs. When taking the union of the sets of conditionally relevant SNPs for the exposure and the outcome, is there a risk that some of the candidate IVs are highly correlated? E.g. at correlation greater than 0.8? In general, what was the maximum correlation between SNPs selected using GCTA-COJO in the data examples? Is the method quite robust to using highly correlated candidate IVs?”*

Reply: Thank you for pointing out this point! You are right that it is possible that SNPs in \mathcal{I}_X and SNPs in \mathcal{I}_Y may be highly correlated, and therefore we did exclude such proteins in the real data analysis using a default COJO cutoff of `--cojo-collinear 0.9` (Line 718). Furthermore, as suggested, in the first set of simulation studies, we have now added a simulation setup with $\rho = 0.8$ in the autoregressive LD structure. We found that the proposed method performed well under this simulation setup, while other methods showed inferior performance compared to weak or moderate LD correlations (Figures 2 and 3).

3. *“Outcome-associated SNPs. The authors are correct to highlight that omitting SNPs that are associated with the outcome could bias the evidence from valid IVs in a linear model of joint SNP effects (if the valid IVs are correlated with the invalid IVs). The recommendation by the authors is then that SNPs that are associated with the outcome should be included. To me, this is not an obvious conclusion. This choice may involve considerable trade-offs that should be investigated:
(a) on whether the plurality condition is more or less likely to hold: the authors suggest the plurality condition is less likely to hold if SNPs associated with the outcome are not considered as candidate IVs (p.23, lns.444–449). If there are a set of candidate SNPs only relevant for the exposure (lets call them C1) that are uncorrelated with many invalid IVs only relevant for the outcome (lets call them C2),*

could the inclusion of these many invalid IVs sway the balance of valid versus invalid candidate IVs? Even if the set of SNPs in C1 were weakly associated with SNPs in C2, would the method in practice give more biased estimates when considering the union of C1 and C2 as candidate IVs, rather than just C1?

Reply: Thank you for the comment! We have added a new simulation setup to reflect the scenario where the SNPs in \mathcal{I}_X (C1 in your comment) and SNPs in \mathcal{I}_Y (C2 in your comment) were uncorrelated or only weakly correlated. Specifically, we simulated 10 SNPs with 5 SNPs in \mathcal{I}_X and 5 SNPs in \mathcal{I}_Y . SNPs *within* \mathcal{I}_X and \mathcal{I}_Y were correlated with an autoregressive structure of $\rho = 0.6$ respectively, while SNPs in \mathcal{I}_X and SNPs in \mathcal{I}_Y were uncorrelated ($\rho_{XY} = 0$) or weakly correlated ($\rho_{XY} = 0.1$). And we applied cisMR-cML using SNPs in $\mathcal{I}_X \cup \mathcal{I}_Y$, and applied cisMR-cML-X, GIVW-X, GEgger-X and LEgger-X using SNPs in \mathcal{I}_X . As shown in Supplementary Table S13, when $\rho_{XY} = 0$, methods only using SNPs in \mathcal{I}_X had unbiased estimates and well-controlled type-I errors as expected. cisMR-cML using all SNPs in $\mathcal{I}_X \cup \mathcal{I}_Y$ also performed similarly well, as it could select out SNPs in \mathcal{I}_Y as invalid. However, when $\rho_{XY} = 0.1$, cisMR-cML-X only using SNPs in \mathcal{I}_X had inflated type-I error and biased estimates, while cisMR-cML still had unbiased estimates and well-controlled type-I error. This new simulation setup again highlighted the importance of including SNPs in \mathcal{I}_Y in the proposed cisMR-cML approach.

The new simulation setup is now described in Lines 634-641, and the results are discussed in the main text Lines 217-223, and the detailed results are given in Supplementary Table S13.

(b) on a “winner’s curse” bias: if candidate IVs are selected based on their measured associations with the outcome, would this compromise the ability to test a null hypothesis of a zero causal effect? There may be an issue in terms of a winner’s curse-type bias. That is, due to sampling uncertainty in the measured genetic associations, retaining SNPs based on how strongly they are associated with the outcome may bias the evidence towards concluding a non-zero causal effect. The current sim-

ulation setting with 10 SNPs may not reflect this, but under a more realistic setup that selects from a large number of correlated SNPs based on their p -values (as in the real data example), I wonder if this trade-off from including outcome-associated SNPs may be more apparent.

Reply: Thank you for the suggestion! First, we would like to clarify that, including SNPs in \mathcal{I}_Y mainly helps to avoid scenarios where all IVs in \mathcal{I}_X become invalid if failing to account for their LD with SNPs in \mathcal{I}_Y . cisMR-cML is expected to select out SNPs in \mathcal{I}_Y as invalid IVs, and the final causal estimate does not depend on the selected invalid IVs, and hence winner's curse due to the inflation of IV-outcome association may be minimal. For example, as illustrated in Supplementary Table S7, in the first set of simulations scenario 2, cisMR-cML-BIC was able to select around 1.9 out of all 2 invalid IVs in $\mathcal{I}_Y \setminus \mathcal{I}_X$ on average across 500 simulation replicates. On the other hand, cisMR-cML-X had highly inflated type-I errors and biased estimates.

To reflect a more realistic scenario of IV-selection as in the real data application, we have conducted a new set of simulation studies (now the second set of simulations in the main text). We started by generating exposure and outcome data from the UK Biobank white-ancestry individual genotype in the *cis*-region of a given protein, then performed exposure and outcome GWAS on all SNPs in the *cis*-region using two non-overlapping samples. We then used a third non-overlapping sample as the reference LD panel and applied GCTA-COJO on the exposure and the outcome GWAS to select SNPs jointly associated with X and jointly associated with Y respectively. We considered two scenarios: in scenario 1, 50 proteins with at least five SNPs associated with the exposure and one SNP associated with the outcome based on the COJO analysis in the proteome-wide application were randomly selected. Then we specified SNPs in \mathcal{I}_X and \mathcal{I}_Y and corresponding genetic effect sizes exactly based on the real-data COJO results and real pQTL/GWAS data. In scenario 2, 50 proteins were randomly selected, and SNPs in \mathcal{I}_X and \mathcal{I}_Y and their corresponding genetic effect sizes were generated randomly. Simulations were repeated 10 times per gene, with a total of 500 replicates. As shown in Figure 4 and Supplementary Table S14, cisMR-cML-X, which only used

SNPs associated with X had inflated type-I error and larger bias and RMSE than cisMR-cML, which used both SNPs associated with X and SNPs associated with Y selected by COJO.

In summary, we believe that the impact of winner's curse in cisMR-cML is less severe than the problem arising from the violation of the plurality assumption due to the omission of adjustments for SNPs associated with the outcome.

The setup of this new set of simulations is now described in Section 4.9, and the results are discussed in the main text Lines 225-253, and the detailed results are given in Supplementary Table S14.

(c) on the number of SNPs used to fit a joint model: including outcome-associated SNPs would increase the regressors in the joint SNP model (Equations (1) and (2) in Section 4.1). Is there a concern that, with a large number of correlated candidate IVs, the co-variance matrices (Σ_X and Σ_Y) in Equation (9) may be more difficult to estimate? Would the p-value threshold of 5×10^{-6} for GCTA-COJO that is recommended by the authors (p.13 lns.263–264) be expected to lead to a manageable number of SNPs to fit this joint model?"

Reply: Thank you for the comment. We agree with you that, in general, with a larger number of SNPs, the genetic correlation/covariance matrix is more likely to be ill-conditioned as pointed out by Reviewer 1. However, we didn't encounter such an issue in our experience so far. This was mainly because, in the proposed approach, we first used COJO to select *jointly* associated SNPs with the exposure or the outcome, which would significantly reduce the number of SNPs compared to using *marginally* associated SNPs. For example, the number of SNPs used in the proteome-wide application ranged from 3 to 20 with a mean of around 5. Nonetheless, we have now added an initial checking step in our R package to warn against an ill-conditioned correlation matrix.

4. *"LD misspecification. As the authors note (p.6 lns.102–103) the LD matrix is often estimated from a publicly available reference panel, and two-sample studies*

are vulnerable to misspecified or mismeasured LD estimates. An extended simulation study should explore sensitivity of the proposed method to misspecification/mismeasurement in LD estimates. Also, is an autoregressive LD structure used in simulations realistic for typical cis-MR studies? If GCTA-COJO is the preferred choice to select candidate IVs, then are there concerns regarding the stability of the results with poor LD estimates compared with a clumping approach that offers some control over how correlated candidate IVs are permitted to be?"

Reply: Thank you for the great suggestion! Yes, we have now added a new set of simulation studies to reflect a more realistic scenario close to our real data application. Specifically, we started by generating exposure and outcome data from the UK Biobank white-ancestry individual genotype in the *cis*-region of a given protein, then performed exposure and outcome GWAS on two non-overlapping samples. We then used a third non-overlapping sample as the reference LD panel and applied GCTA-COJO on the exposure and the outcome GWAS to select SNPs jointly associated with X and jointly associated with Y respectively. The LD matrix used in all examined methods was also estimated using this third non-overlapping sample. To cover a wide range of LD patterns, we considered two scenarios: in scenario 1, 50 proteins with at least five SNPs associated with the exposure and one SNP associated with the outcome based on the COJO analysis in the proteome-wide application were randomly selected. Then we specified SNPs in \mathcal{I}_X and \mathcal{I}_Y and corresponding genetic effect sizes exactly based on the real-data COJO results and real pQTL/GWAS data. In scenario 2, 50 proteins were randomly selected, and SNPs in \mathcal{I}_X and \mathcal{I}_Y and their corresponding genetic effect sizes were generated randomly. Simulations were repeated 10 times per gene, with a total of 500 replicates.

Then we investigated the same implementations of cisMR-cML, LEgger, GIVW-X and GEgger-X as those in our real data analysis. Among the methods assessed, cisMR-cML was the only method that effectively controlled the type-I error and yielded the smallest RMSE. In contrast, LEgger demonstrated generally low power, GIVW-X and GEgger-X using only exposure-associated SNPs in the analysis had unsatisfactory performance

with inflated type-I errors. Detailed results of all examined methods are provided in Supplementary Table S14.

Finally, we agree that poor LD estimates (e.g. due to a mismatched ancestry reference panel) could impact the performance of methods relying on an external LD reference panel, including our proposed method. Due to the nature of the proposed cisMR-cML, which models the **conditional/joint** genetic effects on the exposure and the outcome, we used COJO instead of a standard LD-clumping to select IVs. The clumping approach still requires an LD reference panel which cannot avoid the poor LD estimates issue if exists. More importantly, clumping is based on the marginal genetic association. In the *cis*-MR context, many marginal signals may be driven by the same SNP, and when modeled in the conditional framework, their effect sizes may shrink to zero, potentially introducing more noise into the proposed approach. We have added this comment in the Discussion section (Lines 390-393).

The setup of this new set of simulations is now described in Section 4.9, and the results are discussed in the main text Lines 225-253, and the detailed results are given in Supplementary Table S14.

5. *“Slightly invalid IVs. Does the proposed method rely on a clear separation between what a valid and what an invalid IV is? In practice, this separation may not always be clear, and some SNPs may be slightly invalid IVs. This is usually modeled by local misspecification, where the direct IV effects on the outcome are “local to zero” (decreasing at the same rate as sampling errors $1/\sqrt{n}$). In this case where some IVs are slightly invalid, would the proposed method based on the plurality rule select all slightly invalid IVs? Would this harm inferences (confidence intervals/hypothesis tests)?”*

Reply: Thank you for the suggestion! We have now added a new simulation setup based on the original simulation scenario (1) with $m = 10$ SNPs, $\rho = 0.6$ and $K_1 = 4$ invalid IVs, but the direct effect r_i was generated from $\mathcal{N}(0, (1/\sqrt{N_Y})^2)$, and we fixed the exposure sample size $N_X = 10000$ and varied the outcome sample size $N_Y \in$

{5e4, 1e5, 5e5}. We found that in this case of weakly invalid IVs, it was difficult for the proposed cisMR-cML (based on BIC selection) to select all slightly invalid IVs correctly. For each simulation setup, cisMR-cML-BIC rarely selected any invalid IV (only correctly selected 0.03 ~ 0.1 out of the 4 slightly invalid IVs we have simulated on average). On the other hand, in the original simulation setup, cisMR-cML-BIC correctly selected over 3.5 out of the 4 invalid IVs on average. However, the presence of weakly invalid IVs had little impact on the performance of the proposed method in the simulation. For example, as shown in Supplementary Table S12, it still yielded well-controlled type-I error, while other methods such as GIVW and GEgger had slightly inflated type-I errors. The new simulation setup is now described in Lines 628-633, and the results are discussed in the main text Lines 210-217, and the detailed results are given in Supplementary Table S12.

6. *“Power gain through correlated IVs. The premise of the proposed method is to use correlated IVs to improve power. e.g. p.3 ln.47 discusses the “possibly severe loss of power due to only one or few independent SNPs remaining”. Has this been demonstrated in simulation or by existing work? i.e. given that many SNPs in a single gene region are likely to be highly correlated, is there a significant increase in power from using correlated IVs?”*

Reply: Thanks for the good question. First, from a theoretical point of view, as discussed in Section 1.1 in Burgess et al.¹, incorporating all SNPs jointly associated with the exposure will give us the optimally efficient MR analysis. Second, from a practical viewpoint, many existing robust polygenic MR methods would require at least three IVs in the analysis. Using independent SNPs will often make them inapplicable in the context of cis-MR analysis. Finally and more importantly, only using independent SNPs associated with the exposure will again suffer from the issue of introducing the pleiotropic effect via LD.

We have illustrated the above three points in the additional simulation studies. In the first set of simulation studies, we included the implementation of IVW, Egger,

and MR-cML with independent IVs, referred to as IVW-IND, Egger-IND and cML-IND respectively. As shown in the Figure 2A, in the absence of invalid IV ($K_1 = 0$), all methods using independent IVs had well-controlled type-I errors and almost unbiased estimates. But the power was lower than their correlated-IV counterparts. In particular, compared to GIVW, the power of IVW-IND decreased, as the number of available independent IVs decreased (as ρ increased). Furthermore, when $\rho = 0.6$ or 0.8 , the number of independent IVs was less than three, and thus Egger-IND and cML-IND cannot be applied. As shown in the Figure 2B, in the presence of invalid IVs ($K_1 = 4$), the three approaches using independent IVs had inflated type-I errors because the direct effects of invalid IVs were absorbed in the marginal GWAS effects used in the analysis. In the second set of simulation studies based on real data analysis, we observed a similar type-I error inflation for IVW-IND.

In the second set of simulation studies, we also included the implementation of IVW-IND by using independent IVs marginally associated with the exposure after the standard LD clumping performed in PLINK. As shown in Figure 4, IVW-IND had highly inflated type-I error compared to other examined methods, which again highlighted the importance of the consideration of the third point discussed above.

The above new simulation studies are discussed in the main text Lines 161-163, 167-169, and 225-253, and the detailed results are given in Supplementary Tables S1, S2 and S14.

7. *“Weak IVs. How does the proposed method perform if only weak IVs are available? On p.1 ln.12, the proposed method is described as “robust to violation of any of the three valid IV assumptions”. To me, this would imply that the method offers robustness to a weak IV problem where no strong IVs are available. Is the approach robust to weak IVs or are they selected out?”*

Reply: Thanks for the comment. Yes, our proposed method is robust to violation of any of the three IV assumptions. First, we would like to point out that, in scenario (2) of the original simulation studies (now the first set of simulation studies), where among the

10 SNPs used in the cisMR-cML analysis, 4 SNPs only had effects on X (and thus valid), 1 SNP had an effect on both X and Y (violating no-horizontal pleiotropy assumption), 3 SNPs had effects neither on X nor on Y (violating the relevance assumption), and 2 SNPs only had effects on Y (violating both assumptions), the proposed method still performed well. And as discussed in Supplementary Section S1, the proposed method will only select invalid IVs with $r_i \neq 0$ (violating the second and the third IV assumptions stated in the main text), but not the invalid IVs only violating the first IV assumption ($b_{Xi} = r_i = 0$). However, this will not harm the statistical inference for the causal parameter. As a follow-up analysis, we also tried different implementations of cisMR-cML, and found that using SNPs in $\mathcal{I}_X \cup \mathcal{I}_Y$ gave higher power than mistakenly including some irrelevant IVs (Lines 160-168 in the original main text, now Lines 196-204; and Supplementary Section S3.5).

We have furthermore added two new simulation setups to investigate the weak-IV issue as suggested. Specifically, in the first set of simulations under the scenario (1) with $K_1 = 0$ and $\rho = 0.6$, we reduced either the exposure sample size N_X or reduced the IV-exposure strength (controlled by the variance of X explained by the causal SNPs h_X^2) to generate the weak-IV scenario. When N_X was reduced from 10000 to $\{5000, 1000, 500\}$, the corresponding F -statistics were 27.3, 6.3 and 3.6 respectively; and when h_X^2 was reduced from 0.05 to $\{0.01, 0.005\}$, the corresponding F -statistics were 11.1 and 6.0 respectively. As shown in the Supplementary Tables S10 and S11, when $\theta = 0$, all methods performed well. But when $\theta = 0.2$, cisMR-cML was the only method that yielded unbiased estimates, while other methods including GIVW, GEgger and LEgger had more downward bias (towards the null) as the sample size decreased (and thus more severe the weak-IV issue was). The other four newly-added MR methods MRLDP, MRCorr2, MRCUE and MRAID also had unstable performance.

The new simulation setup is now described in Lines 621-627, and the results are discussed in the main text Lines 205-210, and the detailed results are given in Supplementary Tables S10-11.

8. *“Type I error rates. For the simulation results, I was not able to find the nominated significance level. e.g. are these the results corresponding to a 5% level test? If so, the type I error rates of cisMR-cML under the null hypothesis of no causal effect appear to be quite a bit lower than the nominal level of 5% in Tables 1 and 2. While an under-rejection issue under the null hypothesis is less of a problem compared to larger type I error rates, if the rejection rates under the null hypothesis are much lower than the nominated significance level, this may suggest underlying problems with the theoretical results or code. Could the authors please clarify what the nominated significance level is. If it is 5%, some clarification on this potential under-rejection issue would be useful.”*

Reply: Thank you for the comment. Yes, the results corresponded to a 5% level test, and we have now pointed out this in Lines 153-154.

We agree that cisMR-cML was conservative, especially in the absence of invalid IV. We would like to clarify that, the current (and default) implementation of cisMR-cML uses the data perturbation scheme. Simply using the BIC selection (without data perturbation) may give us inflated type-I error in the presence of invalid IVs for finite sample size problems, and the data perturbation approach is proposed to account for the uncertainty in model selection (i.e. the selection of invalid IVs). As shown in Supplementary Table S1, in the absence of invalid IV, cisMR-cML-BIC yielded correct coverage rate, but when invalid IVs were present (Supplementary Table S2), cisMR-cML-BIC had inflated type-I error. In simulation studies, we found a larger estimated standard error (average across 500 replicates) than the standard deviation of causal estimates (across 500 replicates) for cisMR-cML, resulting in a conservative type-I error rate. This conservative feature of the data perturbation implementation has also been observed before in the independent-IV version of MR-cML⁵ and MVMR-cML³.

We have now acknowledged this conservative feature as a potential limitation in the Discussion section (Lines 400-403), which will be investigated systematically in future studies.

9. *“Comparator methods. The comparator methods considered in simulation are quite limited, making it difficult to understand whether the advantages of the cisMR-cML method are due to robustness to many invalid IVs, conditional/joint modeling of SNP effects, or because methods such as IVW are generally less powerful and less robust than likelihood-based approaches. The authors cite a number of recently developed methods on p.4 ln.57–58 that are not considered as comparator methods in simulation. Some of them, for example, MR-Corr2, also model the joint effects of SNPs. While an exhaustive simulation study of all available methods is not feasible nor the objective, the authors could also consider in simulation a few more state-of-the-art approaches to allow a clearer illustration of the relative merits and drawbacks of the proposed approach.”*

Reply: Thank you for the suggestion! We have now added four methods in our simulation studies, namely MR.LDP, MR.Corr2, MR.CUE and MRAID. We didn't include RBMR as we encountered some bugs when running the R function `RBMR_func`.

We would like to first point out that, although these four methods can handle correlations among the SNPs used in the analysis, they are originally proposed in the polygenic-MR context, where the exposure is polygenic, and SNPs across the whole genome that are associated with the exposure are used in the analysis. However in the *cis*-MR context, we usually focus on one specific LD region, and clearly the exposure such as a gene's or a protein's expression is not as polygenic as other complex traits like BMI. And the commonly used approaches for *cis*-MR analysis are still GIVW-X and GEgger-X. Due to a possibly small number of candidate IVs, these four methods may become unstable. For example, this is discussed as the first limitation in the MRAID paper Discussion section⁶. In our experience, we also found that the trace plots of MR.Corr2 and MR.CUE were sometimes flat at zero, and they could be very sensitive to the hyperparameters used in their priors. In both the first and second sets of simulation studies investigated in the paper, these four methods had inferior performance than the proposed cisMR-cML. We relegate the detailed results for these four polygenic MR methods to the Supplementary to avoid confusion as they were not specifically proposed

for *cis*-MR analysis.

10. “p.3 ln.42, please check grammar on “(as potential drug target)”.”

Reply: Thank you for the comment! We have corrected the sentence and several other typos.

11. “ p.4 ln.50, “only few *cis*-MR methods are robust to the violation of the IV assumptions.” seems at odds with the sentence in the same paragraph “There are several recently proposed methods to account for both LD and horizontal pleiotropy, such as MR-LDP, MR-Corr, MR-CUE, MRAID and RBMR”.”

Reply: Thank you for the comment! We have now added a comment after this sentence in Lines 68-71 “Furthermore, these methods are proposed in the context of using a complex trait as the exposure (or a polygenic MR setup), which uses a relatively large number of SNPs on the whole-genome as IVs, and may not be best suited for the *cis*-MR application, as will be demonstrated later in our simulation studies.”

References

- [1] Stephen Burgess, Verena Zuber, Elsa Valdes-Marquez, Benjamin B Sun, and Jemma C Hopewell. Mendelian randomization with fine-mapped genetic data: choosing from large numbers of correlated instrumental variables. *Genetic epidemiology*, 41(8):714–725, 2017.
- [2] Samuel A Lambert, Arttu Jolma, Laura F Campitelli, Pratyush K Das, Yimeng Yin, Mihai Albu, Xiaoting Chen, Jussi Taipale, Timothy R Hughes, and Matthew T Weirauch. The human transcription factors. *Cell*, 172(4):650–665, 2018.
- [3] Zhaotong Lin, Haoran Xue, and Wei Pan. Robust multivariable mendelian randomization based on constrained maximum likelihood. *The American Journal of Human Genetics*, 110(4):592–605, 2023.

- [4] Jessica MB Rees, Angela M Wood, Frank Dudbridge, and Stephen Burgess. Robust methods in mendelian randomization via penalization of heterogeneous causal estimates. *PloS one*, 14(9):e0222362, 2019.
- [5] Haoran Xue, Xiaotong Shen, and Wei Pan. Constrained maximum likelihood-based mendelian randomization robust to both correlated and uncorrelated pleiotropic effects. *The American Journal of Human Genetics*, 108(7):1251–1269, 2021.
- [6] Zhongshang Yuan, Lu Liu, Ping Guo, Ran Yan, Fuzhong Xue, and Xiang Zhou. Likelihood-based mendelian randomization analysis with automated instrument selection and horizontal pleiotropic modeling. *Science Advances*, 8(9):eab15744, 2022.

REVIEWER COMMENTS

Reviewer #1 (Remarks to the Author):

I am happy with the authors' response to my previous comments and recommend that their manuscript is accepted for publication.

Reviewer #1 (Remarks on code availability):

I tried to download the authors' R package from the GitHub page and was able to do so without any difficulties. I also tried to reproduce the results in their vignette (https://rpubs.com/ZhaotongL/cisMRcML_vignette). The code is reasonably well documented and runs without errors, but the results I got were slightly different to those reported by the authors in the vignette page. This could be due to me using more recent versions of the various R packages - the authors acknowledge that this may produce slightly different results in their vignette. The differences were small enough to not be a cause of concern (e.g. for point estimates in the LDL-CAD analysis, I obtained 0.1834 while the authors reported 0.1846). I have not tried to reproduce their simulation results.

Reviewer #3 (Remarks to the Author):

Review of revised “A robust cis-Mendelian randomization method with application to drug target discovery”

I would like to thank the authors for discussing all of my comments. I think the revision offers more clarity on the strengths and weaknesses of the proposed method. I have only a couple of further minor comments from the previous discussion.

1. **On the reply to point 1 (suitability of the plurality condition for cis-MR).** The authors refer to Sections 4.5 and 4.6, but this does not seem relevant to the point. Both those sections discuss how to accurately assess whether the plurality condition holds. I think for readers looking to apply the method, it is important for them to be able to reason how the plausibility of the assumption may differ in a cis-gene context. The question I had in mind was – what is the value of knowing that the plurality condition holds in a cis-gene context where instruments are correlated?

As far as I understand, the value of the plurality condition in genome-wide MR analyses is to hedge your bets on instrument validity when several instruments provide (somewhat) independent pieces of evidence. In cis-MR with possibly only a few correlated causal variants in the gene region, it would seem that a method based on the plurality rule would be *relatively* less valuable. This does not mean the method is not useful – and the simulation study and example given on transcription factors show that it is indeed not "all or nothing" regarding instrument validity – but I think a slightly more nuanced point/limitation would be useful to note for practical guidance.

2. **On the reply to point 5 (slightly invalid IVs).** I appreciate the extra simulation scenario considered by the authors regarding this point, but generating direct effects as $r_i \sim N(0, 1/N_Y)$ does not provide much insight on the scenario I was suggesting. This describes more of a random-effects type model, and since the asymptotic bias across instruments has mean zero, it's unsurprising that the simulation performance would be good because confidence intervals do not need to be recentered. Instead, a more interesting discussion could consider a fixed-effects type model, where the *mean* of the direct effects are non-zero but decreasing with the sample size at rate $1/\sqrt{N_Y}$, e.g. by setting $r_i = \kappa/\sqrt{N_Y}$ for some non-zero constant κ . This latter specification may induce a $O(1/\sqrt{N_Y})$ order asymptotic bias, and if so, confidence intervals would not be centered correctly based on Theorem 1, even though the estimates would be consistent.

This is a question on the theoretical properties of the method, but with practical relevance because there may not always be a clear distinction between valid and invalid instruments. In this limit experiment, does the method consider these slightly invalid instruments to be valid under the plurality rule assumption, or is the method able to differentiate these slightly invalid

instruments from the truly valid instruments? I wasn't quite able to follow the author response to this comment, e.g. "it was difficult for the proposed cisMR-cML (based on BIC selection) to select all slightly invalid IVs correctly". Does this mean the slightly invalid IVs were selected or not selected? Moreover, is it desirable from the perspective of inference if the method does include (or omit) slightly invalid instruments?

Authors' one-to-one responses

We appreciate the insightful, constructive and helpful comments from the two reviewers. Accordingly, we have made concerted efforts to carefully address each question raised as detailed below. The new or modified parts are highlighted in blue in the main text and Supplementary.

1 Reviewer 1

"I am happy with the authors' response to my previous comments and recommend that their manuscript is accepted for publication."

Reply: Thank you for your positive feedback!

2 Reviewer 3

"I would like to thank the authors for discussing all of my comments. I think the revision offers more clarity on the strengths and weaknesses of the proposed method. I have only a couple of further minor comments from the previous discussion."

Reply: Thank you for your positive feedback! We have made every effort to address the remaining comments you raised in the following point-by-point responses.

1. *"On the reply to point 1 (suitability of the plurality condition for cis-MR). The authors refer to Sections 4.5 and 4.6, but this does not seem relevant to the point. Both those sections discuss how to accurately assess whether the plurality condition holds. I think for readers looking to apply the method, it is important for them to be able to reason how the plausibility of the assumption may differ in a cis-gene context. The question I had in mind was – what is the value of knowing that the plurality condition holds in a cis-gene context where instruments are correlated? As far as I understand, the value of the plurality condition in genome-wide MR analyses is to hedge your bets on instrument validity when several instruments pro-*

vide (somewhat) independent pieces of evidence. In cis-MR with possibly only a few correlated causal variants in the gene region, it would seem that a method based on the plurality rule would be relatively less valuable. This does not mean the method is not useful – and the simulation study and example given on transcription factors show that it is indeed not "all or nothing" regarding instrument validity – but I think a slightly more nuanced point/limitation would be useful to note for practical guidance."

Reply: Thank you for the comment! We agree that in polygenic/genome-wide MR, where independent IVs may represent independent pieces of evidence, the plurality condition is more likely to hold as you suggested. In cis-MR analysis, as LD-correlated (causal) IVs may still represent different mechanisms/evidence, the plurality condition is still reasonable when special action is taken as discussed in Sections 4.5 and 4.6.

In the second application, after accounting for LD-induced pleiotropy, 30 protein-CAD pairs had one invalid IV detected by cisMR-cML-BIC. It is also noted that with a finite sample size, the selection of invalid IVs based on BIC may not be perfect, especially in the presence of weak invalid IVs. With data perturbation, cisMR-cML has detected 310 protein-CAD pairs with at least one invalid IV over 10% of the time during the 100 data perturbations (Lines 299-306).

We have also added a paragraph in the Discussion section to discuss some unique challenges with cis-MR analysis (Lines 402-409).

2. *“On the reply to point 5 (slightly invalid IVs). I appreciate the extra simulation scenario considered by the authors regarding this point, but generating direct effects as $r_i \sim N(0, 1/N_Y)$ does not provide much insight on the scenario I was suggesting. This describes more of a random-effects type model, and since the asymptotic bias across instruments has mean zero, it’s unsurprising that the simulation performance would be good because confidence intervals do not need to be recentered. Instead, a more interesting discussion could consider a fixed-effects type model, where the mean of the direct effects are non-zero but decreasing with the*

sample size at rate $1/\sqrt{N_Y}$, e.g. by setting $r_i = \kappa/\sqrt{N_Y}$ for some non-zero constant κ . This latter specification may induce a $O(1/\sqrt{N_Y})$ order asymptotic bias, and if so, confidence intervals would not be centered correctly based on Theorem 1, even though the estimates would be consistent.

This is a question on the theoretical properties of the method, but with practical relevance because there may not always be a clear distinction between valid and invalid instruments. In this limit experiment, does the method consider these slightly invalid instruments to be valid under the plurality rule assumption, or is the method able to differentiate these slightly invalid instruments from the truly valid instruments? I wasn't quite able to follow the author response to this comment, e.g. "it was difficult for the proposed cisMR-cML (based on BIC selection) to select all slightly invalid IVs correctly". Does this mean the slightly invalid IVs were selected or not selected? Moreover, is it desirable from the perspective of inference if the method does include (or omit) slightly invalid instruments?

Reply: Thank you for the great suggestion! In our previous simulation where $r_i \sim \mathcal{N}(0, 1/N_Y)$, these weak invalid IVs were not selected by cisMR-cML-BIC, and omitting these invalid IVs, cisMR-cML-BIC had inflated type-I error based on asymptotic inference. However, we observed that the default data perturbation version can still control the type-I error.

Per your suggestion, we have conducted new simulations by generating $r_i = \kappa/\sqrt{N_Y}$, and we fixed the exposure sample size $N_X = 10000$ and varied the outcome sample size $N_Y \in \{5e4, 1e5, 5e5\}$ and $\kappa \in \{1, 5, 20\}$. We observed that as κ increased, cisMR-cML-BIC had improved performance in selecting out invalid IVs based on BIC. When $\kappa = 1$, cisMR-cML-BIC barely selected any invalid IVs and had inflated type-I errors; however, cisMR-cML-DP (with data perturbation) still performed reasonably well with controlled type-I errors and nearly unbiased estimates (Supplementary Table S12). When $\kappa = 5$, cisMR-cML-BIC often failed to select out all 4 invalid IVs, and cisMR-cML-DP also yielded biased estimates and slightly inflated type-I errors (Supplementary Table S13). When κ became large ($\kappa = 20$), cisMR-cML-BIC often successfully identified

all 4 invalid IVs and had well-controlled type-I errors (Supplementary Table S14). The new simulation setup is now described in Lines 655-660, and the results are discussed in Lines 213-226.

We also explored the performance of cisMR-cML-BIC based on asymptotic inference when both N_X and N_Y increased and r_i diminished at different rates. In general, the proposed BIC can consistently select invalid IVs (Lemma 1 in Supplementary) as long as r_i does not diminish with the sample size at a rate equal to or faster than $1/\sqrt{N}$. For example, when $r_i = \kappa/\sqrt[3]{N}$, cisMR-cML-BIC consistently selected invalid IVs as N increased, and yielded correct inference (Supplementary Table S16). When $r_i = \kappa/\sqrt{N}$, the selection performance did not improve even as N increased, and the coverage (based on asymptotic inference) was anti-conservative (Supplementary Table S17). When $r_i = \kappa/N$, cisMR-cML-BIC rarely selected any invalid IVs; nevertheless, including such weak invalid IVs did not significantly affect the inference (Supplementary Table S18).

REVIEWERS' COMMENTS

Reviewer #3 (Remarks to the Author):

I thank the authors for addressing my concerns; I have no further questions or comments.

Authors' one-to-one responses

1 Reviewer 3

"I thank the authors for addressing my concerns; I have no further questions or comments."

Reply: Thank you for your positive feedback!